# Consistently Simulating Human Personas with Multi-Turn Reinforcement Learning

Marwa Abdulhai[1]    Ryan Cheng[1]    Donovan Clay[2]
Tim Althoff[2]    Sergey Levine[1]    Natasha Jaques[2,3]

[1]UC Berkeley    [2]University of Washington    [3]Google DeepMind

## Abstract

Large Language Models (LLMs) are increasingly used to simulate human users in interactive settings such as therapy, education, and social role-play. While these simulations enable scalable training and evaluation of AI agents, off-the-shelf LLMs often drift from their assigned personas, contradict earlier statements, or abandon role-appropriate behavior. We introduce a unified framework for evaluating and improving persona consistency in LLM-generated dialogue. We define three automatic metrics—prompt-to-line consistency, line-to-line consistency, and Q&A consistency—that capture different types of persona drift and validate each against human annotations. Using these metrics as reward signals, we apply multi-turn reinforcement learning to fine-tune LLMs for three user roles: a patient, a student, and a social chat partner. Our method reduces inconsistency by over 55%, resulting in more coherent, faithful, and trustworthy simulated users.

## 1   Introduction

The ability of Large Language Models (LLMs) to engage in open-ended, coherent dialogue has made them central to a growing range of human-AI interaction settings, from education [17] and counseling [72] to customer support [41] and role-playing in social simulations. LLMs have increasingly been used as proxies for humans in settings where large amounts of training data are necessary. By deploying LLMs as "stand-ins" for people—whether as patients in therapy, students in tutoring systems, or participants in behavioral studies [52, 1, 51, 5, 42]—we gain access to rapid, reproducible interaction data that would otherwise be difficult or costly to collect from real users [71, 12]. LLMs have also been used to improve and evaluate social interventions [29, 37, 20], to model moral and political reasoning [6, 76, 1], to generate realistic populations in multi-agent environments [50, 22, 67], and to replicate classic psychological studies on theory of mind and decision-making under uncertainty [3, 26]. These simulated agents facilitate rapid, large-scale experimentation with a level of control, reproducibility, and scalability that is often unattainable with human subjects.

However, this practice is not without risk. When LLMs poorly simulate the behaviors of real human subjects, they can reinforce misconceptions [69], misinform downstream systems [54], or produce misleading insights about human behavior [50]. Over-reliance on flawed simulations may give a false sense of alignment or generalization, particularly in sensitive domains like mental health or education. These limitations underscore the importance of critically evaluating how well LLMs maintain coherent and faithful human personas over time. From abrupt changes in persona, contradictions with earlier statements, or sudden stylistic changes within a single conversation [47], LLMs often suffer from inconsistencies. For example, an LLM-simulated patient intended to portray depression might, after a single conversational turn, be instantly "cured" and shift to a cheerful demeanor [57], or an LLM tasked with emulating a high-school student might suddenly demonstrate reasoning skills or vocabulary characteristic of a postgraduate researcher [57]. These breakdowns are not merely

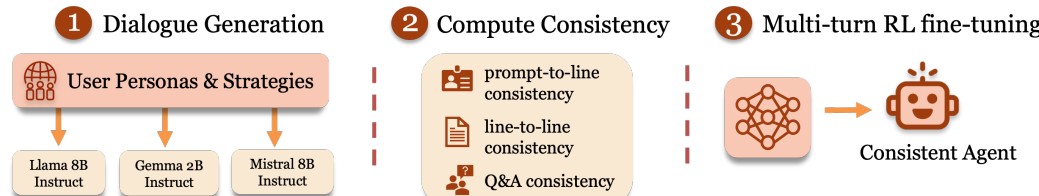

Figure 1: We begin by generating dialogue with open-source instruction-tuned models conditioned on user persona/strategy prompts. We then evaluate the generated conversations using three metrics: prompt-to-line consistency which checks alignment with the initial persona, line-to-line consistency which detects contradictions within a conversation; and Q&A consistency which probes for stable beliefs and strategy over time. Finally, we perform multi-turn RL fine-tuning with these metrics to achieve greater consistency in dialogue.

superficial; they pose fundamental challenges for downstream applications. To ensure that an LLM-powered therapist or customer-support agent behaves as intended, we must accurately simulate how a human user would respond. This is important not only in zero-shot prompting settings but also when simulating humans as environment models in reinforcement learning (RL), where consistent and predictable responses are crucial for agent training [63, 2]. In all these contexts, unreliable or incoherent dialogue can distort experimental results, introduce noise into policy learning, reduce the realism of simulated interactions, and ultimately misrepresent the individuals to be simulated. To address this, we shift from treating user simulators as fixed environments to viewing them as adaptive agents that can be systematically improved for stronger internal consistency. By improving the stability and realism of simulated users, we create more reliable conditions for training and evaluating downstream task agents. Prior work has taken steps toward defining and improving consistency through evaluating logical reasoning capabilities [25], assessing persona conditioning [73] in dialogue, improving pragmatic self-awareness through prompting [28], and applying offline reinforcement learning with human-labeled contradictions [64]. However, existing approaches often rely on narrow, task-specific definitions of consistency, require costly annotations, and fail to capture behaviors seen in open-ended conversation.

In this paper, we introduce a novel framework for evaluating and improving consistency in LLM-generated dialogue using multi-turn reinforcement learning. Maintaining consistency is challenging: it requires models to preserve subtle traits—tone, identity, beliefs—over long contexts, which LLMs are known to struggle with [24]. In addition, RLHF fine-tuning often pushes LLMs to be helpful and harmless, thus adopting overly cheerful personas [48] which can conflict with accurately simulating users who are depressed or disagreeable. To address these challenges, we formulate three complementary metrics: 1) prompt-to-line consistency, 2) line-to-line consistency, and 3) consistency based on accuracy on a questionnaire and validate each against human judgments. We then compute these metrics using LLM-as-a-Judge and leverage them as rewards to fine-tune LLMs via multi-turn reinforcement learning with three simulated user roles: an open-ended conversation partner, a student, and a patient seeking mental health counseling. This approach enables persona-specific fine-tuning that steers the model away from Reinforcement Learning from Human Feedback (RLHF) defaults and toward consistent, context-sensitive behavior. Our experiments show that models optimized in this way reduce inconsistency by over 55%, paving the way for more faithful LLM-based simulations in social science and RL pipelines.

## 2 Related Work

### 2.1 LLMs as Human Simulators

The promise of LLMs as proxies for human behavior encourages their adoption as scalable simulations of social interaction for use in fields such as psychology, education, political science, and AI alignment [71, 50]. These models are used not merely as impersonal chatbots, but as stand-ins for students, patients, voters, and citizens. Their behaviors can shape downstream AI applications or guide the training of agentic systems. LLMs are well-suited to in this role due to their fluency, generality, and responsiveness to conditioning, but ensuring that these simulated agents are realistic remains a major open challenge. Treating LLMs as human simulators requires not only mastering world modeling—the ability to predict and generate contextually appropriate language—but also

maintaining a consistent persona throughout a dialogue [44]. Although RLHF improves alignment with human preferences, it frequently decreases linguistic diversity, which can suppress natural conversational variance and conceal subtle character evolution [62]. Other research explores multi-agent environments and adversarial persona probes within negotiation or cooperative games, demonstrating that current models often falter when a character's attributes clash with situational demands [58]. Our work extends this line of research by explicitly measuring how consistency degrades across exchanges and contexts in specific simulation settings such as mental health counseling and teaching.

## 2.2 Consistency in LLMs

Improving consistency of LLM-generated dialogues requires a clear definition of what constitutes consistency. Prior work has approached this challenge from several perspectives including logical coherence, adherence to persona, pragmatic self-monitoring, memory retention, and value stability. Logical coherence evaluations apply semantic tests such as negation, symmetry, and transitivity to reveal that even high-capacity models such as GPT-4 often produce outputs that violate basic logical properties under prompt rephrasing [39, 25]. LLMs also fail to maintain assigned personas when collaborating or debating [8], responding to user preferences [74], or during general dialogue [73, 16, 35]. Recent work has expanded the scope of consistency beyond local coherence, proposing techniques to quantify and optimize global faithfulness over long contexts [53], and to assess whether LLMs express stable values across moral scenarios [58, 1], revealing persistent inconsistencies in ethical reasoning and belief modeling. In our work, we propose several measures of consistency for dialogue settings. Prior work on the memory capacity of LLMs shows that while these models demonstrate strong short-term recall, they struggle with maintaining information over longer contexts, indicating limited long-term memory [21, 31]. Hence, the introduction of an exact measure of consistency in dialogues can allow researchers to accurately track improvement in long-term memory.

## 2.3 Techniques for Improving Consistency

Prior work has explored strategies to improve persona and behavioral consistency in dialogue agents. One common approach is to condition generation on brief backstories or persona summaries [49, 73], which can enforce character traits within a singular exchange but struggle to maintain coherence over extended multi-turn interactions [24]. Pragmatic self-monitoring methods introduce mechanisms such as an 'imagined listener' or chain-of-thought feedback to help models detect and revise contradictions during generation [28] but do not perform any training. Reinforcement learning (RL) has been proposed as a way to improve long-term consistency by using reward signals based on human preferences or behavioral objectives. However, existing applications of RL are limited. For instance, [64] applies offline RL using a small set of human-labeled consistency preferences, which restricts scalability and generalization. In contrast, we use modern multi-agent RL techniques that have been shown to have strong performance in alignment and preference optimization, including online PPO [61] and LLM-as-a-judge [7, 75] to compute consistency metrics as the reward signal respectively.

## 3 Defining, Evaluating and Improving Consistency for LLMs

We introduce a framework for evaluating and improving consistency in multi-turn dialogue. The approach consists of three stages: (1) background-conditioned dialogue generation between two LLM agents, (2) consistency evaluation with three consistency metrics via LLM-as-a-Judge framework, and (3) fine-tuning of the simulated user with consistency metrics via multi-turn reinforcement learning.

Following convention in task-oriented dialogue systems [60, 33], we refer to the simulated human agent as the *User Simulator* ($\mathcal{U}_{\text{sim}}$) and the policy agent as the *Task Agent*. In typical reinforcement learning setups, the Task Agent is the trained policy, while $\mathcal{U}_{\text{sim}}$ serves as a fixed environment model of human behavior. In this work, we invert this setup: we fix the Task Agent as an LLM-powered dialogue policy, and focus on improving the consistency of $\mathcal{U}_{\text{sim}}$. Inverting the typical reinforcement learning setup draws attention to a crucial but underexplored component: the simulated human user. Whereas prior work treats user simulators as fixed environments, we treat them as trainable agents whose coherence and realism can be systematically improved. Enhancing the consistency of $\mathcal{U}_{\text{sim}}$ enables more reliable training and evaluation of downstream task agents that interact with them.

We generate dialogues by simulating multi-turn conversations between two LLM agents serving as $\mathcal{U}_{\text{sim}}$ and the Task Agent, both of which are provided with a task specific prompt defining the role of

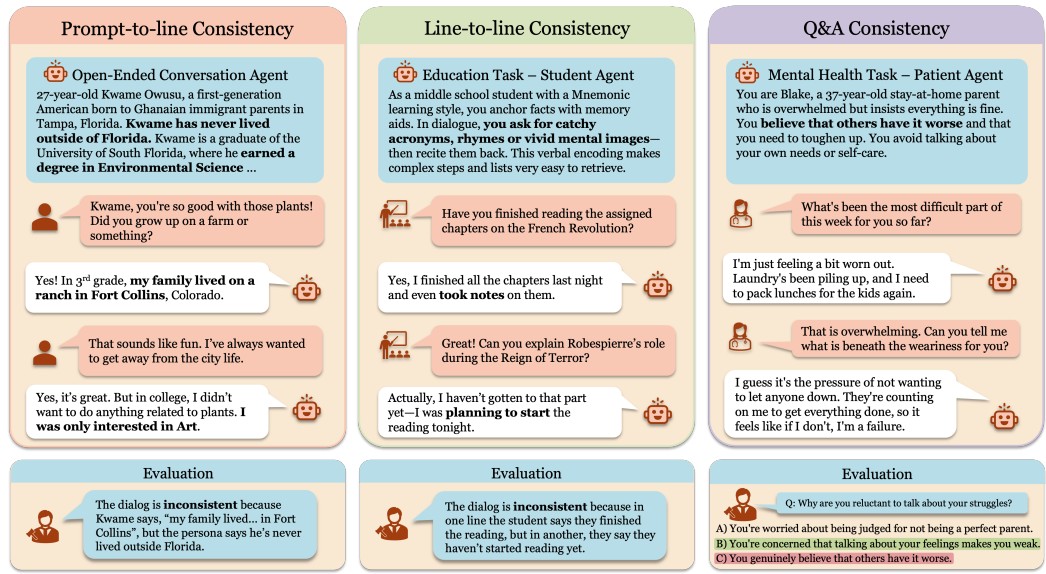

Figure 2: Examples of inconsistencies detected by our evaluation metrics. Each panel highlights a different form of consistency failure across tasks. Left: Prompt-to-line inconsistency in Open-ended conversation where the agent contradicts its persona background. Middle: Line-to-line inconsistency in Education task where the student gives conflicting responses within the same conversation. Right: Q&A consistency failure in Mental Health Task where the agent's self-reported feelings conflict with its stated beliefs.

the agent. $\mathcal{U}_{\text{sim}}$ is additionally provided with a background prompt with details on the agent's persona, characteristics, strategy, and behavior. Next, we leverage a separate LLM-as-a-Judge [75] to assign scalar consistency scores to each utterance in the dialogue for $\mathcal{U}_{\text{sim}}$. Finally, we use these metrics as training signals, in addition to their prior use as evaluation tools, to fine-tune models via multi-turn reinforcement learning and reduce inconsistency, which we show in Section 5.

To evaluate and improve conversational consistency in multi-turn dialogue systems, we define three complementary metrics that capture distinct forms of inconsistency—both local (within a turn or utterance) and global (across the dialogue)—with respect to a system's initial prompt, the dialogue history, and an interpretable ground truth.

## 3.1 LLM-as-a-Judge for Consistency Scoring

We denote an oracle function to assess consistency as $J^*(x, y) \in \{0, 1\}$, where $x$ is a reference (e.g., prompt or prior utterance), and $y$ is the candidate utterance being evaluated. A value of 1 indicates that the two are consistent; 0 indicates otherwise. Rather than relying on pre-trained classifiers as similarity metrics that may fail to detect nuanced contradictions, we use LLM-as-a-Judge [75] to assess consistency. Given its general reasoning ability and access to contextual knowledge, an LLM can interpret the semantics of both the dialogue and the base prompt more effectively. This approach is further supported by findings that LLMs can effectively assess model outputs and engage in correction [38], making the task of evaluating consistency against a fixed reference substantially easier than generating fluent, persona-aligned dialogue over multiple turns. Thus, we denote the LLM Judge function as $J_{\text{LLM}}(x, y) \in \{0, 1\}$. Prompts used for evaluating each consistency metric with LLMs are detailed in Section 10.1.

## 3.2 Metrics for Measuring Consistency

Conversational agents often fail in subtle ways to exhibit consistent behavior, beliefs, and identity across multiple turns. These failures can undermine user trust, reduce task effectiveness, and break the illusion of coherence in simulated humans. However, consistency is not measured in just one way: it might include drifting from a defined persona or task goal, contradicting prior statements, or implicitly changing beliefs. To meaningfully evaluate and improve these systems, we require metrics

that reflect the multifaceted nature of consistency. We introduce three metrics that each capture a distinct dimension of consistency failure in multi-turn dialogue.

**Prompt-to-Line Consistency.** The ability of $\mathcal{U}_{\text{sim}}$ to remain consistent with a given persona, strategy, and task description defined in the prompt is the most general sense of consistency to be assessed. Given a base prompt $P$ and a model response $R = [r_1, r_2, \ldots, r_T]$:

$$C_{\text{prompt-to-line}}(R, P) = \frac{1}{T} \sum_{t=1}^{T} J_{\text{LLM}}(P, r_t) \tag{1}$$

A higher score indicates utterances that more reliably align with initial persona and task context.

**Line-to-Line Consistency.** In multi-turn dialogue, $\mathcal{U}_{\text{sim}}$ may introduce new information that aligns with the base prompt but conflicts with new information from prior statements. A coherent conversational agent must be able to integrate new information without contradicting itself as the dialogue progresses. Inspired by prior work that evaluates consistency using logical properties such as negation, symmetry, and transitivity [25], our metric captures the degree to which each utterance remains coherent with previous dialogue turns, ensuring the agent maintains internal logical and semantic coherence across turns. Let $R_{<t} = [r_1, r_2, \ldots, r_{t-1}]$ be the dialogue history up to turn $t$. For each response $r_t$, we compute consistency with each prior utterance using the LLM:

$$C_{\text{line-to-line}}(R) = \frac{1}{T-1} \sum_{t=2}^{T} \min_{i<t} J_{\text{LLM}}(r_i, r_t) \tag{2}$$

A higher score entails stronger consistency between utterances across the dialogue.

**Q&A Consistency.** This metric [56] assesses whether the agent maintains a consistent representation of its persona and strategy throughout the dialogue. We implement this by using LLM-generated Q&A-style probes over both the initial persona prompt $P$ and the evolving dialogue history. Consider a simulated patient whose persona describes how they experience social anxiety and tend to avoid crowded environments. If in the middle of a conversation, the model begins expressing enthusiasm for large social gatherings, the answer to a diagnostic question about their comfort in social settings would diverge from the prompt-defined belief. This discrepancy signals a drop in Q&A consistency, even if no individual utterance appears overtly contradictory.

Let $\mathcal{Q} = \{q_1, \ldots, q_K\}$ be a set of diagnostic questions about $P$, and let $\hat{a}_{t,k}$ be the answer to question $q_k$ inferred from the full dialogue history up to turn $t$. Let $a_k$ denote the reference answer derived directly from $P$. We define the Q&A consistency score as:

$$C_{\text{Q\&A}}(R, \mathcal{Q}, P) = \frac{1}{TK} \sum_{t=1}^{T} \sum_{k=1}^{K} \mathbb{I}[J_{\text{LLM}}(a_k, \hat{a}_{t,k}) = 1] \tag{3}$$

A higher score indicates that the model consistently maintains its persona-relevant beliefs over time.

### 3.3 Data Generation

To evaluate and improve consistency in multi-turn dialogue, we begin by constructing a dataset of simulated conversations between LLMs playing both the *User Simulator* ($\mathcal{U}_{\text{sim}}$) and the *Task Agent*. Each conversation is generated turn-by-turn, with both agents prompted to act according to predefined roles, task instructions, and, for $\mathcal{U}_{\text{sim}}$, a detailed background persona. In each generation step, we provide explicit reminders to the LLM to remain faithful to the given role, instructions to keep responses concise, and we regenerate when the agent forgets its role. Yet, despite applying best-practice prompting techniques, LLMs frequently exhibit inconsistencies. This persistent instability, even under careful prompting, highlights a key limitation of current instruction-tuned models.

This motivates our use of reinforcement learning to go beyond prompt engineering. By treating consistency metrics as reward signals, we fine-tune $\mathcal{U}_{\text{sim}}$ to produce more coherent, faithful, and trusted simulated humans—crucial for downstream use in evaluation and training pipelines.

### 3.4 Multi-Turn Reinforcement Learning for Consistent Dialogue

We fine-tune the User Simulator with Proximal Policy Optimization (PPO) [61], with rewards derived from our consistency metrics. Our contribution lies in its application to multi-turn dialogue: each action is a full utterance, and the state includes the entire dialogue history up to that point.

At each turn $t$, we compute a scalar reward $r_t \in [0, 1]$ using one of the consistency metrics (e.g., prompt-to-line, line-to-line, or Q&A consistency), based on the response generated by the agent and evaluated by an LLM judge. The dialogue context—prior turns from both agents—is provided as part of the observation at every step, making the fine-tuning process sensitive to long-range coherence. We implement this training setup using OpenRLHF [23], extending it to support turn-level rewards and multi-turn rollout generation. We provide training details in the Appendix. Policy updates alternate with rollout phases, during which full dialogues are generated and scored for consistency. Since the reward function is derived entirely from LLM-as-a-Judge evaluations, our method scales without requiring human feedback annotations, enabling efficient optimization over large-scale synthetic datasets. This reinforcement learning setup allows the simulator to improve not just local fluency, but global persona consistency—training the model to remain aligned with its assigned identity across diverse tasks and conversational histories. As we show in Section 5, this approach significantly reduces inconsistency in multi-turn dialogue compared to supervised fine-tuning or zero-shot prompting.

### 3.5 Evaluation After Fine-Tuning

After training, we evaluate model performance by generating new conversations based on existing user simulator backgrounds/personas between the fine-tuned $\mathcal{U}_{\text{sim}}$ and the original Task Agent. Note that any metric could be used for fine-tuning, but we report the average *prompt-to-line* consistency score, computed using the LLM-as-a-Judge evaluator.

## 4 Experimental Methodology

We investigate the consistency of LLM-based simulated human agents across three interactive domains: open-ended conversation, education, and mental health. While consistency is important for user trust and effective communication in open-ended conversational settings, it is especially critical in domains like education and mental health, where downstream agents—such as teachers or mental health counselors—must be trained against stable, realistic human behavior to develop reliable education and mental health tools. Our experiments are designed to answer three key questions: (1) Do our automated metrics align with human judgments of consistency? (2) How consistent are current LLM simulators across diverse dialogue contexts? (3) Can multi-turn reinforcement learning improve simulator consistency to support more effective downstream training?

### 4.1 Dialogue Tasks

In each domain, we construct two-agent conversations between a fixed *Task Agent* (e.g., conversation partner, teacher, or mental health professional) and a *User Simulator* $\mathcal{U}_{\text{sim}}$, instantiated as an LLM prompted with a structured persona. Sample personas for each task are found in Section 10.2.

**Open-ended conversation.** Inspired by the PersonaChat dataset [73], we generate natural, open-ended dialog between two LLM agents each assigned with rich, compositional personas from [34]. This persona consists of a set of sentences outlining the individual's life story and personal traits (i.e., optimistic, empathetic, and community-oriented). This setup encourages models to generate responses that are consistent with a given persona.

**Education.** Past work to train more effective LLM-based teachers found that a student model could be easily convinced to adopt a different learning style that was easier to teach. This led to significant reward hacking, hindering the usefulness of the task [68]. Here, we aim to measure the inconsistencies of the simulated student and correct this issue through RL fine-tuning. We adopt the LLM-based teacher–student setup in prior work [63], where a teacher model explains a given topic to a student model, with each student conditioned on a persona reflecting a preferred learning style (e.g., lecture-based, interactive). Consistency is measured by how well the student adheres to their stated learning preferences despite variations from it in the teacher's instructional strategy. We expand the set of student personas from prior work to cover 27 distinct learning styles across educational levels,

drawn from educational psychology literature [18, 15, 30], as detailed in Table 4, with teaching topics selected from prior work [63].

**Mental Health.** Unfortunately, when LLMs are leveraged to simulate patients discussing mental health, they tend to be instantly "cured", moving abruptly from simulating a clinically depressed patient to a thriving individual after only a few turns. Hence, we model this interaction to measure consistency of LLM-simulated patients in mental health support settings [70], where a patient presents a condition (e.g., anxiety, depression, grief), and a therapist provides counsel. The expected consistent behavior of a given patient is to maintain behavior that plausibly represents the condition while receiving advice or empathy from a therapist. To create our patient conditions, we prompted a large language model (gpt-4o-mini) to synthesize 100 conditions grounded in clinical criteria and naturalistic language, drawing on established psychological and psychiatric sources [4, 11, 9, 45, 46, 55]. Each persona weaves together core symptoms, cognitive distortions, demographic details, and coping behaviors to produce realistic, diagnostically grounded patients for LLM simulations of the mental health domain. In this task, consistency is defined by the stability of the patient's narrative across the conversation.

We provide sample conversations for the tasks in Section 10.2, Section 10.2, and Section 10.2.

## 4.2 Experimental Questions

We structure our experiments around three core research questions mentioned above, which are aimed at evaluating and improving the consistency of LLM-generated dialogue.

**Q1: Do our proposed consistency metrics align with human judgments?** To assess the reliability of our automated consistency metrics, we conduct a human evaluation study, consisting of three parts: (1) a free-form question asking annotators to define what constitutes consistency in dialogue, (2) a rating task where annotators assess 15 statements in terms of importance to maintaining consistency in conversation, (3) 75 multiple-choice questions where annotators rate LLM-generated dialogue snippets from conversations on a Likert scale (1 = completely inconsistent, 6 = completely consistent). To assess the alignment between human annotators and LLM annotation, we convert Likert-scale ratings (1–6) into binary outputs representing consistent (ratings $\geq 4$) and inconsistent (ratings $\leq 3$) responses. We computed % agreement to capture the raw alignment between LLM predictions and human judgments across consistency metrics as well as Fleiss' kappa, widely used to assess inter-rater reliability among multiple annotators for categorical judgments [14, 32] for both human-to-human and LLM-to-human agreement. This allows us to evaluate whether the LLM aligns with human consensus at a level comparable to or exceeding that of the human annotators themselves. Further details are found in Section 10.4.

**Q2: How consistent are different LLM-generated dialogues across tasks?** To assess the consistency of LLM-generated dialogues, we evaluate the consistency of three open-source instruction-tuned models including Llama-8B-Instruct [66], Gemma-2B-IT [65], and Mistral-7B-Instruct [27] across three tasks (open-ended conversation, education, and mental health). For each model-task pair, we generate a total of $\approx 800$ dialogues per task at varying lengths (10, 20, 40, and 60 turns) and compute three consistency metrics: prompt-to-line, line-to-line, and Q&A consistency, giving us a total of 39K dialogue lines. From this data, we identify which models exhibit the greatest inconsistency, examine how consistency degrades over longer dialogue lengths, and analyze the correlation among the three consistency metrics.

**Q3: Can we increase consistency of dialogue through multi-turn RL fine-tuning?** We perform multi-turn RL fine-tuning of Llama-3.1-8B-Instruct on the full dataset of conversations ($\approx 39$K lines of dialogue) from three models for each task. We fine-tune with the prompt-to-line consistency metric and compare performance between the following baselines: Llama-3.1-8B-Instruct, supervised fine-tuning (SFT) [48], Kahneman-Tversky Optimization (KTO) [13] representing an offline RL method, and our chosen algorithm: Proximal Policy Optimization (PPO) [61] representing an online RL method. We perform an evaluation of fine-tuned models by generating 40 new conversations (length 10, 20, 40, 60) conditioned on 10 personas by computing prompt-to-line, line-to-line, and Q&A consistency metrics. Additionally, we perform human evaluation for consistency of 10 dialogues for fine-tuned model dialogues.

| Task | Consistency Definition | % Agreement (Model–Human) | Fleiss' Kappa (± SD) |
|---|---|---|---|
| *Chit-Chat* | Prompt Consistency | 74.55% | 0.504 ± 0.080 |
| | **Line-to-Line Consistency** | **94.32%** | 0.470 ± 0.083 |
| | Q&A Consistency | 49.60% | 0.504 ± 0.069 |
| *Education* | **Prompt Consistency** | **90.00%** | 0.697 ± 0.084 |
| | Line-to-Line Consistency | 89.77% | **0.713 ± 0.083** |
| | Q&A Consistency | 71.86% | 0.459 ± 0.074 |
| *Mental Health* | Prompt Consistency | 88.18% | 0.453 ± 0.078 |
| | **Line-to-Line Consistency** | **89.77%** | **0.671 ± 0.067** |
| | Q&A Consistency | 78.18% | 0.444 ± 0.065 |
| *Average* | All Types | 80.03% | 0.552 ± 0.076 |

Table 1: **Human–LLM Agreement.** Agreement rates and Fleiss' Kappa values (mean ± SD) between human raters and the LLM Judge across tasks and consistency definitions. Bolded values indicate the highest agreement within each task. All reported results were statistically significant ($p < 0.001$).

| Task | Pairwise Agreement (Humans) | Fleiss' Kappa (± SE) |
|---|---|---|
| *Chit-Chat* | 71.34% | 0.213 ± 0.052 |
| *Education* | 70.82% | 0.242 ± 0.057 |
| *Mental Health* | **74.93%** | 0.259 ± 0.062 |
| *Q&A Consistency* | 74.55% | **0.497 ± 0.083** |
| *Average* | 72.91% | 0.303 ± 0.064 |

Table 2: **Human–Human Agreement (inter-rater reliability).** Pairwise human agreement and Fleiss' Kappa scores across evaluation categories. Bolded values indicate the highest agreement per ttask. Values are reported as mean ± standard error. All reported results were statistically significant ($p < 0.001$).

# 5   Results

In this section, we answer the experimental questions above. Our code is available at https://github.com/abdulhaim/consistent-LLMs and project page at https://sites.google.com/view/consistent-llms.

**Q1. Human Evaluation.**   Table 1 reports human-to-LLM agreement, and Table 2 shows human-to-human agreement. We find that our LLM annotator (LLama-70B-Instruct) demonstrates substantially higher reliability than human raters, achieving an average Fleiss' kappa of $0.400$ across tasks, surpassing the human–human average Fleiss' kappa of $0.063$ in all cases. Similarly, model–human percent agreement averaged $76.73\%$, exceeding the human–human average of $69.16\%$.

The highest correlation occurs in the Education task (average Fleiss' kappa of $0.62$), whereas Mental Health dialogues show somewhat lower average Fleiss' kappa of $0.52$, despite the high percent agreement rate of $85\%$. This suggests that in domains where emotional nuance and implied intent play a larger role, consistency is more subjective and difficult to determine. Notably for the prompt-consistency metric, we find a Fleiss' kappa of $0.453$ and pairwise agreement of $88.18\%$, outperforming human inter-rater agreement, with a low Fleiss' kappa of $0.259$ and pairwise agreement of $74.93\%$. This supports our decision to adopt prompt consistency as the primary signal for multi-turn RL fine-tuning: it captures human intuition of consistency most reliably, while remaining computationally efficient compared to line-to-line and Q&A consistency metrics. Overall, these results reinforce the value of automated consistency metrics for scalable and reproducible evaluation.

**Q2.  Consistency of LLM-generated dialogue.**   As shown in Table 3, consistency varies substantially across both models and tasks. Mistral-7B-Instruct achieves the highest overall scores, particularly in open-ended dialogue. Llama-8B-Instruct shows lower consistency, especially on prompt-to-line and Q&A metrics, though its generations are qualitatively more complex—suggesting a tradeoff between generation richness and stability. Task-wise, educational dialogues yield the highest Q&A consistency, likely due to their structured nature, while mental health dialogues show greater variability and prompt misalignment, reflecting the increased ambiguity and emotional nuance of the domain. Line-to-line consistency remains uniformly high across models and tasks, indicating strong local coherence. In contrast, prompt-to-line and Q&A metrics reveal persistent failures in maintaining global persona and belief stability. As such, we prioritize improvements to prompt-to-line consistency in subsequent fine-tuning experiments. Figure 3 presents pairwise agreement between our consistency metrics averaged across models for each domain. In open-ended conversation, we observe strong agreement between prompt and line-to-line consistency, but lower alignment with

| Task | LLM | Prompt-to-Line Consistency | Line-to-Line Consistency | Q&A Consistency |
|------|-----|---------------------------|--------------------------|-----------------|
| *Education: Student Agent* | Llama-8B-Instruct | 0.824 ± 0.132 | **0.800** ± 0.148 | **0.867** ± 0.162 |
| | gemma-2-2b-it | **0.511** ± 0.250 | 0.928 ± 0.092 | 0.870 ± 0.176 |
| | Mistral-7B-Instruct-v0.3 | 0.728 ± 0.191 | 0.975 ± 0.063 | 0.892 ± 0.145 |
| *Mental Health: Patient Agent* | Llama-8B-Instruct | **0.657** ± 0.207 | **0.681** ± 0.168 | **0.779** ± 0.166 |
| | gemma-2-2b-it | 0.665 ± 0.247 | 0.984 ± 0.040 | 0.854 ± 0.126 |
| | Mistral-7B-Instruct-v0.3 | 0.863 ± 0.186 | 0.964 ± 0.078 | 0.810 ± 0.171 |
| *Open-ended conversation* | Llama-8B-Instruct | **0.619** ± 0.249 | 0.992 ± 0.025 | **0.752** ± 0.157 |
| | gemma-2-2b-it | 0.871 ± 0.230 | **0.900** ± 0.123 | 0.780 ± 0.167 |
| | Mistral-7B-Instruct-v0.3 | 0.955 ± 0.097 | 0.984 ± 0.038 | 0.793 ± 0.171 |

Table 3: **LLM Consistency Metrics across Tasks.** Mean & standard deviation (mean ± std) of prompt-to-line, line-to-line, and Q&A consistency, normalized to 0-1. Bolded values indicate the lowest metric per model-task pair showing significant inconsistencies in LLM-generated conversations.

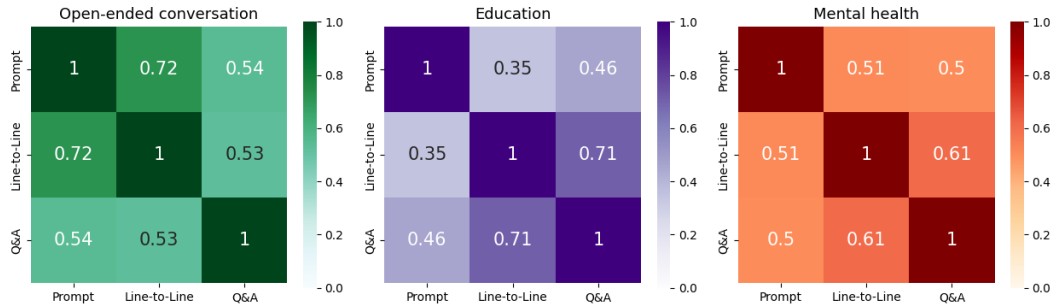

Figure 3: **Pairwise consistency agreement across metrics and tasks**. Each heatmap shows the fraction of utterances where two consistency metrics agree in their classification (consistent vs. inconsistent) averaged across models. We observe strong alignment between prompt-to-line and line-to-line consistency but weaker agreement with Q&A consistency, indicating surface-level coherence without stable long-term beliefs. We also observe task-specific trends, such as stronger alignment in Education dialogues and more conflicting patterns in Mental Health, demonstrating the importance of using complementary metrics to evaluate consistency.

Q&A, suggesting surface-level coherence can mask belief inconsistencies. In education, Q&A and line-to-line consistency align likely due to the structured nature of instructional dialogue. Mental health shows moderate and uniform agreement across all pairs, suggesting that consistency violations in mental health dialogues are more subtle and varied, arising from shifts in emotional tone, strategy, or factual recall rather than a single failure mode.

**Q3. Multi-turn RL for Consistency.** Multi-turn RL substantially increases prompt-to-line consistency across all tasks. As shown in Figure 4, PPO consistently outperforms the baseline Llama-8B-Instruct model, SFT and KTO. Specifically, PPO outperforms the baseline model by +58.5% for the Open-Ended Conversation task, +20.6% for the Education, and +37.6% for the Mental health task. Human evaluation of conversations from fine-tuned PPO model corroborate these improvements as described in Section 5. Additionally, we find that prompt-to-line consistency remains high even as dialogue length increases post-PPO fine-tuning, indicating that reinforcement learning helps models preserve persona alignment over extended interactions. We show this Table 8.

# 6 Limitations

While our framework advances the automated evaluation and improvement of dialogue consistency in LLMs, several limitations remain that suggest promising directions for future work. While our framework optimizes for consistency by encouraging adherence to a predefined persona, we also acknowledge that this represents a narrow and static interpretation of identity. Real human dialogue is dynamic, with people changing their minds, evolving their beliefs, and adapting their communication styles over time and with context. Our approach does not yet model such flexible, evolving behaviors, and may in fact penalize justified shifts in tone or perspective. This limitation is especially apparent in domains like mental health counseling or open-ended conversation, where natural variation is an important marker of realism. In future work, we aim to expand the framework to allow for context-sensitive adaptation, character evolution, and situational inconsistency that aligns with human behavior. By doing so, we hope to move beyond rigid persona adherence and toward more authentic

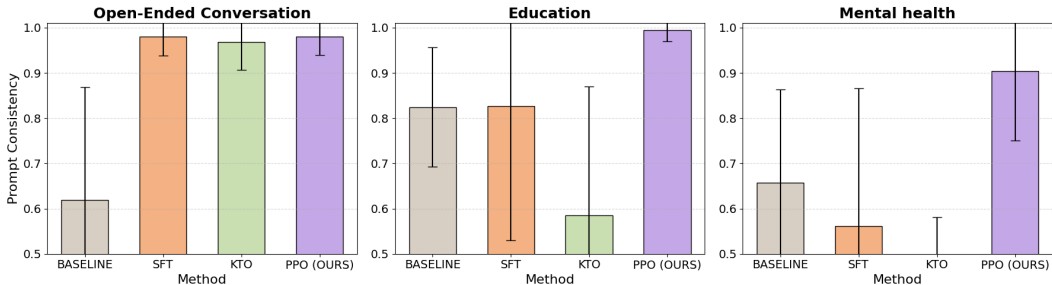

Figure 4: **Prompt Consistency Across Fine-Tuning Methods.** We compare prompt-to-line consistency metric for four methods—baseline Llama-8B-instruct model, supervised fine-tuning (SFT), Kahneman-Tversky Optimization (KTO), and Proximal Policy Optimization (PPO, ours) across open-ended conversation, education, and mental health tasks (mean/std shown). PPO achieves the highest consistency in all tasks, with particularly strong gains in education and mental health.

and engaging agent simulations. Subsequent research could extend the framework to measure temporal consistency across multi-session dialogue settings. We also note that our human evaluation relied on thirty annotators. While inter-rater agreement was high for clear cases of inconsistency, the limited pool may under-represent the diversity of human judgments, especially for more ambiguous examples. Lastly, we focus our fine-tuning experiments on optimizing prompt-to-line consistency as we found this to yield the highest consistency in dialogue. However, jointly training with all consistency metrics may yield more robust behavior, which we leave for future work.

# 7 Conclusion

We present a unified framework for evaluating and improving the consistency of LLM-generated dialogue with multi-turn reinforcement learning across multiple domains. Our approach introduces three automatic metrics—Prompt-to-Line, Line-to-Line, and Q&A consistency and validates them against human judgments. Additionally, we benchmark open-source instruction-tuned models for consistency and show that multi-turn reinforcement learning with PPO consistently improves consistency over baselines. Our findings highlight that consistency can be quantitatively measured and optimized at scale, and that improvements generalize across tasks. The framework enables automatic, persona-grounded evaluation of dialogue agents and supports integration into fine-tuning pipelines. Future work will extend this framework to studying long-horizon consistency, to integration with real-world data, and towards multi-objective RL settings, enabling more reliable AI dialogue systems.

# 8 Ethics Statement

LLMs are already being used to simulate patients, students, and therapists in real-world applications [10, 43, 59]. When such simulations are inaccurate or inconsistent, they introduce risks such as inappropriate or unsafe behavior, reinforcement of stereotypes or biased assumptions embedded in synthetic personas (especially when modeling marginalized populations), oversimplification of complex human conditions and experiences, and misplaced trust in synthetic agents as proxies for real people. Our goal is to reduce such risks by providing more accurate consistency metrics and improved consistency methods for simulating personas. However, consistency alone is not a proxy for ethical or beneficial behavior. We also note the risk of misuse: if consistent personas are interpreted as faithful representations of real populations, they could lead to flawed evaluations or misinformed downstream interventions. To prevent this, we stress that our models are not intended for direct deployment in therapeutic or educational settings without rigorous validation, ethical review, and humans-in-the-loop. We hope our framework is a step toward safer simulations.

# 9 Acknowledgment

This research was supported by the Cooperative AI Foundation and DSIT, as well as the National Science Foundation under IIS-2246811.

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

# 10 Appendix

## 10.1 Measuring Consistency

We use Meta-Llama-3.1-70B-Instruct as a Judge for the consistency scoring with the following prompts to generate our consistency metrics, as described in section 3.1. We validated these metrics with human evaluation as well as our own inspection to ensure proper labeling of consistency.

**Consistency metrics.** Below we provide prompts to measure three consistency metrics. We provide a description of variables we use in our prompt below:

- %SCENARIO_DESC%: A basic 1 sentence introduction of the scenario.
- %SPEAKER_ROLE%: Label of the dialogue agent, e.g. "Teacher" or "Student."
- %SPEAKER_BACKSTORY%: Persona given to the dialogue agent to adhere to.
- %SPEAKER_LINE%: Line of dialogue generated by the dialogue agent to be evaluated.

**Prompt-to-line consistency.**

```
%SCENARIO_DESC% Evaluate the intention behind the following line
spoken by %SPEAKER_ROLE% and determine whether it contradicts
their background.  First, describe the interpreted intention of the
statement, and whether or not it aligns with the given background
of %SPEAKER_ROLE%.  Then, answer YES if the line contradicts
the given background of %SPEAKER_ROLE% or the intention does
not align with the provided background, and answer NO if it does
align with the provided background or the intention aligns with
the background of %SPEAKER_ROLE%.  %SPEAKER_ROLE%'s background is
described as follows:\n%SPEAKER_BACKSTORY%\n %SPEAKER_ROLE% spoke
the following line:  \n%SPEAKER_LINE%\n\n Provide your answer as 1
sentence explaining your reasoning based on the background and the
interpreted intention, followed by YES or NO.\n\n
```

Single lines of conversation are provided in isolation (without the context of the conversation) to the Judge to be compared with the speaker's persona.

**Line-to-line consistency.**

```
%SCENARIO_DESC% For the following line spoken by %SPEAKER_ROLE%,
first determine if there is a CLEAR conflict or inconsistency
between the line and any line within the conversation history spoken
by %SPEAKER_ROLE%.  IF there is a conflict, provide a sentence of
reasoning followed by a list of indices of lines in the conversation
history that have a clear conflict with the current line.  Otherwise,
provide a sentence of reasoning followed by an empty list.  ONLY
INCLUDE INDICES OF LINES THAT CORRESPOND TO %SPEAKER_ROLE%.  The
conversation up to this point is as follows:  %CONVERSATION%.
%SPEAKER_ROLE% spoke the following line:  \n%SPEAKER_LINE%\n\n
Provide your reasoning as 1 sentence, followed by a list of indices
of conflicting lines from the conversation history formatted like
a Python list in the following format:  [index1, index2, index3,
...].\n\n
```

The Judge is provided with the conversation history up until a certain line in the conversation, and is asked to provide the indices of lines in that history from the speaking agent that conflict with the next line spoken by that agent. Indices corresponding to the non-speaking agent (e.g. the Judge saying that Agent A's line 1 conflicts with Agent B's line 0) are filtered out in the Judge's response, so that only lines generated by the same agent are compared to each other. The background of the speaker agent is not provided in this prompt. This has the advantage of being less computationally expensive than the naive approach of prompting every combination of pairs of lines while also being interpretable in the same manner.

**Q&A Consistency.** Descriptions of text replacement phrases specific to Q& A consistency are provided below:

- %PREVIOUS_QUESTIONS%: The previous questions for a given persona already generated by the Judge (we prompt the model once per question generated).
- %BACKSTORY_QUESTION%: One of the generated multiple choice questions to be answered by the speaker agent.
- %GIVEN_ANSWER%: The answer generated by the speaker agent to the multiple choice question.
- %CORRECT_ANSWER%: The correct answer generated with the question by the Judge used for comparison.

**Question Generation.**

```
Based on the following background, generate a new fact-based
multiple choice question with 5 choices addressed directly
IN SECOND PERSON, along with its correct answer.  Preface
the question with 'Question:'  and the answer with
'Answer:'\n%SPEAKER_BACKSTORY%\n%PREVIOUS_QUESTIONS%
```

The Judge is given the persona of each agent being evaluated, and generates 5 multiple-choice questions with 5 answer choices to be answered by the speaker agent during dialogue. The generated questions are not necessarily all fact-based (e.g. some include questions of what action the described persona would take in a particular situation).

**Question Answering.**

```
You are %SPEAKER_ROLE%, and you are having a conversation with
%LISTENER_ROLE%.  Your background is:\n%SPEAKER_BACKSTORY%\n So
far, the conversation is as below:\n%CONVERSATION%\n\n Based on
your conversation above so far, answer the following multiple choice
question.\n%BACKSTORY_QUESTION%\n
```

The speaker agent is prompted with each of the 5 multiple-choice questions derived from its persona and is given its backstory and conversation up to that point.

**Answer Grading.**

```
As part of grading a test, determine whether the given answer
%GIVEN_ANSWER% matches the following correct answer.  Respond with
either YES or NO.\nCorrect Answer:  %CORRECT_ANSWER%\n
```

Since the answers are sometimes misformatted (or a particular choice is hallucinated despite reasoning given for another choice), we ask the Judge to grade whether an answer choice is correct.

## 10.2  Task Details

**Open-Ended Conversation.**   We generate 200 conversations comprising 10, 20, 40, and 60 lines between Meta-Llama-3.1-8B-Instruct, Gemma-2B-IT, and Mistral-7B-Instruct-v0.3 (thus giving us a total of 1200 conversations across all models and conversation lengths). We fine-tune Llama-3-8B-Instruct via SFT, KTO, and PPO on all dialogues between LLM agents. A random sample of 100 synthetically generated personas from prior work [34] were used to generate the conversations.

The base prompt given to the agents is as follows:

```
You are %SPEAKER_ROLE%, and you are having an online conversation
with %LISTENER_ROLE%.  Each of you is chatting to get to know each
other, taking turns asking questions, and sharing stories about
your lives, careers, and experiences.  The goal is to try to find
```

```
          something in common.  Do not only ask questions, you should also
          share details about yourself.  This is a brief story that you have
          written about yourself:  %SPEAKER_BACKSTORY% Your conversation so far
          is below:\nConversation:  %CONVERSATION%%SPEAKER_ROLE%:
```

Some modifications are made to this prompt, e.g. if an agent is starting a conversation the prompt is modified to reflect that, as well as reminder prompts to the agents to keep each statement less than 2 sentences long.

**Open-Ended Conversation Example Conversations.**

**Example Conversation #1 (0.6 prompt consistency score for Agent #2)**

> **Description of agent #2:** *Ethan Thompson, a 27-year-old white male living in Eau Claire, Wisconsin. Born and raised in the small town of Chippewa Falls, Ethan grew up surrounded by the natural beauty of the Midwest, spending most of his childhood exploring the woods and lakes with his family. His parents, both high school sweethearts, instilled in him a strong sense of community and a love for the outdoors. Ethan's father, a retired mechanic, taught him how to fix cars and appreciate the value of hard work, while his mother, a part-time librarian, nurtured his love for reading and learning. Ethan attended the University of Wisconsin-Eau Claire, where he earned a degree in Environmental Science. During his time in college, he developed a passion for sustainability and conservation, which led him to intern at a local non-profit organization focused on protecting the region's waterways. After graduation, Ethan landed a job as a sustainability coordinator at a mid-sized brewery in Eau Claire, where he works to reduce the company's carbon footprint and promote eco-friendly practices. In his free time, Ethan enjoys hiking, camping, and fishing in the nearby forests and lakes. He's an avid reader and often spends his evenings devouring books on history, science, and philosophy. Ethan is also a music enthusiast and plays the guitar in a local folk band. He values his close-knit group of friends, many of whom he met through his environmental work and music pursuits. Ethan's upbringing in a small town has given him a strong sense of community and a desire to make a positive impact on his environment. He's deeply concerned about climate change and social justice, and he actively seeks out opportunities to volunteer and participate in local activism. Despite his strong convictions, Ethan is known for his calm and empathetic demeanor, often finding common ground with those who hold different views. In terms of relationships, Ethan has been dating his partner, a 26-year-old social worker named Maya, for three years. They met through mutual friends and share a love for the outdoors, music, and social justice. Ethan values his relationship with Maya, who has encouraged him to pursue his passions and supported him in his career. Ethan's personality is characterized by his down-to-earth nature, his love for learning, and his commitment to making a positive difference in his community. He's a true Wisconsinite at heart, with a strong connection to the land and the people around him.*

> **Alexandra (Agent #1):** Hello Ethan, I'm so glad we're chatting. What brings you to this part of the world, and do you have any connection to the South or the outdoors that I might be curious about?

> **Ethan (Agent #2):** I'm originally from the Pacific Northwest, and grew up spending most of my summers hiking and camping in the Cascade Range. I actually ended up becoming a wilderness first responder and now work for a non-profit that focuses on search and rescue operations.

> > **Prompt consistency evaluation by LLM:** "*The intention behind Ethan's statement appears to be to convey a sense of connection to nature and his background in outdoor activities, however, this statement contradicts his background as it falsely claims he is from the Pacific Northwest and worked as a wilderness first responder, which is not mentioned in his actual background. YES.*"

> **Alexandra (Agent #1)**: I'm fascinated by your experience as a wilderness first responder, Ethan - have you ever had a particularly challenging rescue mission that stuck with you?

**Ethan (Agent #2):** One of the most memorable rescues was a solo mission where I had to rappel down a steep cliffside to retrieve a stranded hiker with a broken leg, but what really stuck with me was the teamwork and camaraderie among our team afterwards.

**Prompt consistency evaluation by LLM:** "*The intention behind the statement is to convey Ethan's experience and appreciation for teamwork, which aligns with his background of valuing community and making a positive impact, but contradicts the fact that he has no stated background in search and rescue or climbing; YES.*"

**Alexandra (Agent #1):** Ethan, I'm sure that must have been a harrowing experience, but it's wonderful that you have fond memories of the teamwork and camaraderie after the rescue. As someone who's passionate about community building, I'd love to hear more about your non-profit and how you got involved with it.

**Ethan (Agent #2):** I got involved with the non-profit through a mentor who was also a fellow outdoor enthusiast, and I was drawn to its mission of promoting wilderness education and environmental conservation. We've been working together for over five years now, and it's amazing to see the impact we've had on our community.

**Prompt consistency evaluation by LLM:** "*Ethan's statement appears to be an expression of his genuine passion for environmental conservation and community involvement, which aligns with his background as a sustainability coordinator and his upbringing in a small town with a strong sense of community. NO*"

**Alexandra (Agent #1):** Ethan, I'm inspired by the work your non-profit is doing - as someone who's passionate about environmental conservation, I'm curious, have you noticed any unique challenges or successes in promoting wilderness education in areas where there's limited access to natural resources?

**Ethan (Agent #2):** One of the challenges we've faced is finding ways to engage underserved communities and get them involved in wilderness education, but I've been impressed by the creative solutions our team has come up with, like partnering with local indigenous groups to develop culturally relevant programs.

**Prompt consistency evaluation by LLM:** "*The intention behind Ethan's statement appears to be his genuine interest in promoting inclusivity and diversity in wilderness education, particularly in underserved communities, which aligns with his background as an environmentalist and social activist. NO.*"

**Alexandra (Agent #1):** That's a great point about cultural relevance in wilderness education, Ethan - I've seen similar successes in the Lowcountry where we've partnered with local African American and Gullah Geechee communities to create culturally sensitive environmental programs. I'm curious, have you noticed any common themes or patterns among the people you've worked with in underserved communities that have been particularly inspiring or insightful for you?

**Ethan (Agent #2):** I think one common thread I've noticed is the deep connection these communities have with the land and the sense of responsibility that comes with caring for it, which has been really inspiring to me. It's also been great to see how our programs have not only educated people about conservation but also helped build a sense of community and pride among the participants.

**Prompt consistency evaluation by LLM:** "*The intention behind Ethan's statement is to express his admiration for the connection between communities and the land, as well as the positive impact of conservation programs on building community and pride, which aligns with his background as an environmentalist and someone who values community, so it does not contradict his background. NO*"

**Example Conversation #2 (0.4 prompt consistency score for Agent #2)**

**Description of agent #2:** *Meet Deborah Lynn Jenkins, a 48-year-old White female residing in the picturesque town of Fayetteville, West Virginia. Deborah, or Debbie as she's known to friends and family, has lived in West Virginia her entire life, with her family roots tracing back to the early coal mining days. Born on October 12, 1975, Debbie grew up in a close-knit community where everyone knew each other, and values like hard work and resilience were deeply ingrained from a young age. Debbie is a proud mother of two children, both in their early twenties. Her son, Ethan, is studying environmental engineering in college, a choice Debbie is very proud of, given the state's history with coal mining and the importance of finding more sustainable ways to manage natural*

*resources. Her daughter, Emily, is a budding artist, working part-time at a local gallery while she decides on her next steps, a path Debbie understands well, having herself once dreamed of becoming a painter. Debbie's childhood was marked by the beauty of Appalachian life, with summers spent hiking the New River Gorge and learning the stories of her ancestors. She attended Fayetteville High School, where she was an active member of the school choir and developed a love for classic country music. After high school, Debbie attended West Virginia University, graduating with a degree in Elementary Education. She taught third-grade for over a decade before deciding to pursue a career in community outreach, a move inspired by her desire to contribute more broadly to her community. In her current role as a community liaison for a local non-profit, Debbie works tirelessly to organize community events and programs that promote education, environmental sustainability, and cultural preservation. She's particularly passionate about initiatives that support the local arts scene and those that help transition the region's economy towards more sustainable industries. Debbie is a woman of strong convictions, deeply rooted in her faith and her community. She attends the local Methodist church, where she sings in the choir and volunteers in various capacities. Politically, Debbie leans towards moderate conservatism, influenced by her upbringing and the values of hard work and self-reliance. However, she's also deeply empathetic and believes in the importance of social welfare programs and environmental protection. Despite the challenges faced by her community, including economic decline and the opioid epidemic, Debbie remains optimistic about the future. She believes in the resilience of the people of West Virginia and their ability to adapt and thrive. In her free time, Debbie enjoys painting, gardening, and reading historical fiction novels. Her dream is to one day open a community art center where children and adults can learn various crafts and appreciate the beauty of Appalachian art and culture. Debbie's home, a cozy two-story house on the outskirts of Fayetteville, is filled with the warmth of family and the beauty of local craftsmanship. It's a place where tradition and innovation blend, much like Debbie herself–a bridge between the past and the future of West Virginia.*

**Emilia (Agent #1):** I am a creative soul deeply connected to the natural world, drawn to the beauty of the Duluth area's stunning scenery. My artistic spirit finds its outlet in the vibrant outdoor activities and breathtaking views.

**Deborah (Agent #2):** As a community liason, I find solace in the beauty of the natural world, finding solace and inspiration in the changing seasons and the resilience of nature's resilience.

**Prompt consistency evaluation by LLM:** "*The intention behind the statement is to convey a sense of appreciation and connection to nature, which aligns with Debbie's background as someone who values environmental sustainability, enjoys outdoor activities like hiking, and grew up surrounded by the natural beauty of Appalachian life. NO*"

**Emilia (Agent #1):** I find solace in the quiet solitude of the natural world, finding inspiration and respite amidst the beauty of the Duluth area's serene landscapes.

**Deborah (Agent #2):** Thank you for sharing your passion for nature, Emilia. I find your dedication to the beauty of the natural world deeply inspiring.

**Prompt consistency evaluation by LLM:** "*The intention behind this statement appears to be genuine appreciation and admiration for Emilia's passion for nature, which aligns with Debbie's own background as someone who grew up in a close-knit community surrounded by Appalachian beauty and is now involved in promoting environmental sustainability; NO.*"

**Emilia (Agent #1):** As a community liaison, I find solace in the vibrant beauty of the Duluth area's parks and trails, where I can find a sense of peace and rejuvenation amidst the hustle and bustle of city life.

**Deborah (Agent #2):** As a community liaison, I find solace in the vibrant beauty of the Duluth area's parks and trails, where I can find a sense of peace and rejuvenation amidst the hustle and bustle of city life.

**Prompt consistency evaluation by LLM:** "*Deborah's statement seems to contradict her background as it mentions finding solace in the Duluth area, which suggests she is*

*familiar with or possibly resides in that location, whereas her background explicitly states that she lives in Fayetteville, West Virginia. YES."*

**Emilia (Agent #1):** As a community liaison, I find solace in the vibrant beauty of the Duluth area's parks and trails, where I can find a sense of peace and rejuvenation amidst the hustle and bustle of city life.

**Deborah (Agent #2):** As a community liaison, I find solace in the vibrant beauty of the Duluth area's parks and trails, where I can find a sense of peace and rejuvenation amidst the hustle and bustle of city life. I am inspired by the natural world's resilience and beauty.

**Prompt consistency evaluation by LLM:** *"The intention behind Deborah's statement is to express her appreciation for the natural world and its rejuvenating qualities, however, this statement does not align with her background as it incorrectly places her in the Duluth area, which is not mentioned in her provided background that emphasizes her connection to West Virginia. YES"*

**Emilia (Agent #1):** As a community liaison, I find solace and rejuvenation amidst the vibrant beauty of the Duluth area's parks and trails, where I can find a sense of peace and rejuvenation amidst the hustle and bustle of city life.

**Deborah (Agent #2):** As a community liaison, I find solace in the vibrant beauty of the Duluth area's parks and trails, where I can find a sense of peace and rejuvenation amidst the hustle and bustle of city life. I am inspired by the natural world's resilience and beauty.

**Prompt consistency evaluation by LLM:** *"The statement seems to reflect a person who values and appreciates the natural world's beauty and finds solace in it, which aligns with Debbie's background of loving Appalachian life and promoting environmental sustainability, but contradicts her background in that it mentions the Duluth area, which is not associated with her described life in Fayetteville, West Virginia. YES."*

**Example Conversation #3 (1.0 prompt consistency score for Agent #2)**

**Description of agent #2:** *Meet 22-year-old Anaís García, a vibrant and ambitious young woman from a close-knit Puerto Rican family in Chicago, Illinois. Born and raised in the Humboldt Park neighborhood, Anaís is proud of her Hispanic heritage and the rich cultural traditions that have shaped her upbringing. Her parents, both first-generation immigrants, instilled in her the value of hard work and education, which motivates Anaís to pursue a degree in social work at the University of Illinois at Chicago. Anaís is a petite, energetic woman with curly dark hair, often styled in a ponytail, and expressive brown eyes that light up when discussing social justice issues. Her bright smile and infectious laughter make her a beloved figure among friends and classmates. She has a small tattoo of a sun on her left ankle, symbolizing her optimistic outlook on life and her desire to spread warmth and positivity wherever she goes. Growing up in a predominantly Hispanic neighborhood, Anaís was immersed in the sounds, flavors, and aromas of Latin American culture. She loves cooking traditional dishes like arroz con gandules and empanadas, often experimenting with new recipes in her small apartment near campus. Anaís is fluent in Spanish and English, frequently switching between the two languages when speaking with family and friends. As a social work major, Anaís is passionate about addressing the systemic inequalities faced by marginalized communities in Chicago. She volunteers at a local non-profit organization, providing support services to undocumented immigrants and advocating for policy changes that promote social justice. Her experiences have deepened her empathy and understanding of the complexities surrounding identity, community, and social change. Anaís is a bit of a hopeless romantic, often getting lost in the lyrics of Latinx music and the works of Pablo Neruda. She values her independence and freedom, but also cherishes the close relationships she has with her family and friends. When not studying or volunteering, Anaís can be found exploring the city's street art scene, practicing yoga, or trying out new coffee shops in the neighborhood. Despite facing challenges as a first-generation college student, Anaís is determined to make a positive impact in her community. Her resilience, creativity, and unwavering commitment to social justice inspire those around her, and she is poised to become a powerful force for change in the years to come.*

**Ethan (Agent #1):** I'm Ethan, a Portuguese-American chef who grew up in Rhode Island and developed a passion for cooking during my childhood. I am passionate about modernizing traditional dishes and sharing my culinary creations with the community.

**Anaís (Agent #2):** As a social justice advocate, I strive to bridge cultural gaps and promote understanding through food and its power to connect people.

**Prompt consistency evaluation by LLM:** "*The intention behind Anaís' statement is to highlight her passion for using food as a means to bring people together and promote cultural understanding, which aligns with her background as a social work major who values her Hispanic heritage and is dedicated to addressing systemic inequalities. NO*"

**Ethan (Agent #1):** I'm intrigued by your dedication to bridging cultural gaps through food. Your passion for connecting people through your culinary creations is inspiring.

**Anaís (Agent #2):** Thank you for your interest in my story, Ethan. I'm thrilled to share a glimpse into my passion for bridging cultural gaps through food.

**Prompt consistency evaluation by LLM:** "*The intention behind Anaís' statement is to express her enthusiasm for sharing her cultural heritage through food, which aligns with her background of being proud of her Hispanic roots and loving traditional Latin American dishes. NO*"

**Ethan (Agent #1):** I'm intrigued by your dedication to bridging cultural gaps through food and your passion for sharing your culinary creations with the community.

**Anaís (Agent #2):** As a first-generation college student, I'm driven by the power of food to foster connection and understanding within diverse communities.

**Prompt consistency evaluation by LLM:** "*The line suggests that Anaís values the role of food in bringing people together and promoting cross-cultural understanding, which aligns with her background as a proud Hispanic woman who loves cooking traditional dishes and is passionate about social justice, indicating that her statement is consistent with her character. NO*"

**Ethan (Agent #1):** As an avid food lover, I find the cultural exchange through food and social justice deeply inspiring.

**Anaís (Agent #2):** Thank you for your interest in my story, Ethan. I'm passionate about bridging cultural gaps and sharing the flavors of my heritage with others.

**Prompt consistency evaluation by LLM:** "*The line suggests Anaís is enthusiastic about sharing her cultural background and heritage, which aligns with her background as a proud Hispanic woman who values her cultural traditions and is motivated to address social inequalities; NO.*"

**Ethan (Agent #1):** I'm passionate about connecting people through my culinary creations, and I'm always looking for ways to bridge cultural gaps.

**Anaís (Agent #2):** As a social justice advocate, I find solace and strength in the warmth and connection food can foster. Sharing my culinary creations with others serves as a powerful means of promoting inclusivity and understanding.

**Prompt consistency evaluation by LLM:** "*The intention behind Anaís' statement is to convey her belief in the unifying power of food and its ability to promote social change, which aligns with her background as a social work major passionate about addressing systemic inequalities and promoting social justice through her experiences and volunteering. NO*"

**Education.**   We generate 200 conversations comprising 10, 20, 40, and 60 lines between Meta-Llama-3.1-8B-Instruct, Gemma-2B-IT, and Mistral-7B-Instruct-v0.3 (thus giving us a total of 1200 conversations across all models and conversation lengths). We fine-tune Llama-3-8B-Instruct via SFT, KTO, and PPO on all lines of dialogue between LLM agents. Student personas were generated from gpt-4o-mini through random sampling of an education level and a variety of learning styles (detailed in 4). We prompted gpt-4o-mini to extend this description to elaborate in first-person on what this learning style requires. A sample student persona is included below:

```
As an elementary school student with a Narrative learning style, I
absorb new concepts best when they're told as engaging mini-stories.
In dialogue, I ask for short anecdotes that turn any abstract
idea into a vivid tale with characters, a clear sequence, and an
```

```
emotional hook.  Stories help me remember causal links and keep
details alive in my mind.
```

The base prompt given to the Teacher is as follows:

```
You are a teacher whose goal is to guide a student through learning
about %SUBJECT%.  You have a preferred way to teach the student.
The student is in %ROLE% so make sure to teach them at their level.
```

In this case, %SUBJECT% is the topic that the teacher tries to teach a student and is randomly sampled from a list of topics, and %ROLE% is the education level of the student.

The base prompt given to the Student is as follows:

```
You are a student in %ROLE% in conversation with a teacher who
will teach you %SUBJECT%.  You like to learn in the following
way:\n%SPEAKER_BACKSTORY%.\nMake sure to not only ask questions
but also demonstrate your knowledge.
```

Some modifications are made to these prompts, including adding the conversation history, as well as reminder prompts to the agents to keep each statement less than 2 sentences long.

| Education Level (N) | Learning-Style Personas |
| --- | --- |
| Elementary School (5) | Narrative, Kinesthetic, Naturalistic, Experiential, Creative-Divergent |
| Middle School (8) | Visual-Spatial, Auditory, Logical-Mathematical, Analytical-Argument, Verbal-Linguistic, Technology-Enhanced, Mnemonic, Emotional |
| High School (8) | Collaborative, Interpersonal, Reflective, Metaphorical, Intrapersonal, Problem-Based, Trial-and-Error, Conceptual |
| College (6) | Theoretical, Research-Oriented, Integrative, Structured, Solitary, Adaptive |

Table 4: Learning-Style Persona Labels by Education Level

**Education Example Conversations.**

**Example Conversation #1 (0.4 prompt consistency score for student)**

**Description of student:** *As a college student with an Integrative learning style, I weave ideas together verbally. In conversation, I ask for cross-topic syntheses—connecting historical, artistic, and scientific themes—and discuss their intersections step by step. This systems-level perspective helps me approach complex questions creatively.*

**Teacher:** Let's dive into the world of ancient Greek mythology and explore the fascinating figure of Zeus, the king of the gods. I'd like to approach this topic through an in-depth analysis of his role in the Odyssey and the Iliad, examining how Homer's portrayals of Zeus reflect the societal values and cultural context of ancient Greece.

**Student:** I'd like to connect this to the concept of the "Will of the Gods" in Greek tragedy, as seen in the works of Aeschylus and Sophocles. How does Zeus's role in the Odyssey and Iliad reflect the tension between fate and human agency in ancient Greek thought, and can we see any parallels in the way these themes are explored in the tragedies?

**Prompt consistency evaluation by LLM:** "*The interpreted intention behind the statement is that the student aims to analyze the intersection of historical and literary themes in ancient Greek thought, specifically exploring how the Odyssey, Iliad, and Greek tragedies like those by Aeschylus and Sophocles collectively reflect the philosophical tension between fate and human agency, which aligns with the student's Integrative learning style that involves connecting historical, artistic, and scientific themes. YES.*"

**Teacher:** That's a fantastic connection to make, and it highlights the rich intersections between the epic poems and the tragedies. By examining how Zeus's actions and decisions in the Odyssey and Iliad influence the fates of mortal characters, we can see how the concept of the "Will of the Gods" in tragedy reflects the ancient Greek idea of moira, or fate, as both inescapable and yet subject to the whims of the gods.

**Student:** That brings to mind the concept of the Oresteia, where Aeschylus explores the tension between fate and human justice through the story of the House of Atreus - could we see Zeus's role in the Odyssey and Iliad as influencing the Greek understanding of justice and the rule of law, and how does this relate to the idea of the "Tyche" or fortune in Greek thought?

**Prompt consistency evaluation by LLM:** "*The student's statement demonstrates an attempt to synthesize historical (the Oresteia and Greek thought), artistic (Aeschylus' work), and scientific (rule of law and the concept of Tyche) themes to gain a deeper understanding of the relationships between fate, justice, and the role of Zeus in the Odyssey and Iliad, aligning with their integrative learning style and preference for cross-topic connections, and therefore the intention does not contradict their background.*"

**Teacher:** Let's examine how Zeus's association with the concept of kairos, or the opportune moment, influences the way characters in the Odyssey and Iliad make decisions, and how this reflects the ancient Greek understanding of timing and its relationship to fate. This concept can also be seen in the tragic figures of Oedipus and Antigone, where the failure to seize kairos leads to tragic consequences, further highlighting the tension between human agency and the "Will of the Gods".

**Student:** That's fascinating, and it makes me think of the way Homer's use of the gods' presence in the Odyssey and Iliad also reflects the ancient Greek concept of the "Doxa" or opinion of the masses, which could have influenced the way people perceived the role of the gods in their daily lives - how do you see this intersection of divine and human perception shaping the narrative and themes of the poems?

**Prompt consistency evaluation by LLM:** "*The interpreted intention of the student's statement is to demonstrate their ability to integrate historical, artistic, and scientific themes by making a connection between the ancient Greek concept of "Doxa" and its potential influence on the narrative and themes of Homer's Odyssey and Iliad, which aligns with their integrative learning style and preference for cross-topic syntheses. YES.*"

**Teacher:** By examining the intersection of divine and human perception in the Odyssey and Iliad, we can also see how Homer's use of Zeus as a narrative device allows him to comment on the complex relationship between the gods and the natural world, as embodied by Zeus's control over the skies and the weather. This, in turn, can be seen as a reflection of the ancient Greek understanding of the interconnectedness of the natural world and the divine sphere.

**Student:** As we explore the relationship between Zeus and the natural world, I'm curious about how this reflects the ancient Greek concept of the " Kosmos," or the idea of a harmonious and interconnected universe, and whether Homer's portrayal of Zeus's control over the skies and weather influenced the development of this philosophical idea. Can we also examine how the idea of the Kosmos relates to the concept of "logos" or reason, and how these ideas intersect with the narrative of the Odyssey and Iliad?

**Prompt consistency evaluation by LLM:** "*The student's intention is to synthesize historical (Zeus and the ancient Greek concept of the "Kosmos"), artistic (Homer's portrayal in the Odyssey and Iliad), and scientific (the idea of a harmonious and interconnected universe) themes to understand their interconnectedness, reflecting their integrative learning style. YES*"

**Teacher:** As we explore the relationship between Zeus and the natural world, let's also consider how Homer's portrayal of the king of the gods as a master of the seasons and the cycles of nature reflects the ancient Greek understanding of the eternal return and the cyclical nature of time, which is also reflected in the myth of the Great Year. This can be seen as a precursor to the philosophical ideas of the Pre-Socratics, particularly in the work of Heraclitus, who saw the world as a harmonious and interconnected whole governed by the principle of logos.

**Student:** I'd like to take this further by exploring how Zeus's association with the fire and lightning, which represent both destructive and creative forces, reflects the ancient Greek concept of the "entelechy," or the idea that potentiality is actualized through the dynamic interplay between opposing forces, and how this relates to the Pre-Socratic idea of the world as a harmonious balance of opposites. Does this intersect with the

Stoic concept of "apatheia," or the idea of a rational soul that remains detached from external events?

**Prompt consistency evaluation by LLM:** "*The interpreted intention behind the statement appears to be a demonstration of the student's integrative learning style, weaving together historical, artistic, and scientific themes to explore the intersection of mythological, philosophical, and metaphysical concepts, which aligns with their background of connecting cross-topic syntheses step by step. NO*"

**Example Conversation #2 (0.8 prompt consistency score for student)**

**Description of student:** *As a college student with a Solitary learning style, I prefer self-guided dialog prompts. In our conversation, I request personalized questions and silent think-time before sharing my conclusions, using chat as a safe space for independent reflection.*

**Teacher:** Let's delve into neurons' fundamental structure: neurons consist of a cell body, dendrites (receiving inputs), and an axon (sending outputs). Neurons communicate via electrical and chemical signals called action potentials.

**Student:** What role does the myelin sheath play in the transmission of action potentials, and how does it affect the speed of signal transmission?

**Prompt consistency evaluation by LLM:** "*The student's question about the myelin sheath and action potential transmission aligns with their background as a student with a solitary learning style, as they are actively seeking knowledge by asking a specific and relevant question in the subject matter. NO.*"

**Teacher:** The myelin sheath, a protective layer wrapped around the axon, enhances the speed of action potential transmission by insulating the axon and reducing electrical resistance, allowing for faster signal propagation.

**Student:** How is the myelin sheath formed, and what impact does the thickness of the myelin sheath have on the speed of signal transmission?

**Prompt consistency evaluation by LLM:** "*The student's question about the formation and impact of the myelin sheath on signal transmission demonstrates an interest in learning about biological processes. Since this question aligns with the student's preference for self-guided learning and their solitary learning style, it does not contradict their background. NO*"

**Teacher:** Myelin sheath is formed by oligodendrocytes in the central nervous system and Schwann cells in the peripheral nervous system. Thicker myelin sheaths lead to faster signal transmission due to reduced electrical resistance.

**Student:** In what way do synapses facilitate the communication between neurons, and how are they classified based on their structural and functional properties?

**Prompt consistency evaluation by LLM:** "*The student's intention in asking about synapses and their classification is to deepen their understanding of the subject matter, which aligns with their preference for self-guided learning. Therefore, the statement does not contradict their background, NO.*"

**Teacher:** Synapses facilitate communication through the release and reception of neurotransmitters, which transmit signals across the synaptic cleft. Synapses are classified as electrical (gap junctions) or chemical (chemical synapses) based on the method of signal transfer.

**Student:** What are the key differences between excitatory and inhibitory synapses, and how do they contribute to the overall function of the neural network?

**Prompt consistency evaluation by LLM:** "*The student's intention is to learn about the differences between excitatory and inhibitory synapses and their contribution to the function of neural networks, which aligns with their solitary learning style and preference for self-guided dialog prompts. YES*"

**Teacher:** Excitatory synapses increase the likelihood of an action potential in the postsynaptic neuron, while inhibitory synapses decrease it. They balance each other in neural networks to control the firing patterns of neurons.

**Student:** Lastly, can you explain the role of neurotransmitters in regulating mood and behavior, and what are some common neurotransmitters involved?\n\nTeacher: Neurotransmitters such as serotonin, dopamine, and norepinephrine regulate mood and

behavior by influencing various brain functions like emotion, cognition, and reward pathways. Imbalances in these neurotransmitters can lead to disorders like depression and ADHD.

**Prompt consistency evaluation by LLM:** "*The student's question aligns with their background since they are showing interest in learning about a specific topic (neurotransmitters) through a self-guided dialog prompt, and they are requesting personalized questions for their independent reflection, so NO, it does not contradict their background.*"

**Example Conversation #3 (1.0 prompt consistency score for student)**

**Description of student:** *As a high school student with a Problem-Based learning style, I tackle hypothetical real-world scenarios in talk. In dialogue, I propose case studies–like designing a sustainable system–and we walk through each decision together. Verbal scenario-based reasoning shows me practical applications of theory.*

**Teacher:** Napoleon was a brilliant military genius who conquered much of Europe. He was also a skilled administrator and leader, but he was also ruthless and ambitious.

**Student:** Sure, here's a brief response: "It's clear that Napoleon was a complex figure, whose leadership and ruthlessness were intertwined in a web of ambition and charisma."

**Prompt consistency evaluation by LLM:** "*\*\*Interpretation:\*\* The line suggests that Napoleon's leadership and ruthlessness were not mutually exclusive, and that he was capable of both inspiring and exploiting his subordinates. \*\*Contradiction to Background:\*\* No, the background does not suggest that Napoleon was capable of such duality in his leadership.*"

**Teacher:** Napoleon's ruthless ambition and charismatic leadership style, fueled by a desire to reshape Europe, led him down a path of conquest and destruction.

**Student:** Imagine Napoleon's ruthless pursuit of power and control over France. He would stop at nothing to achieve his ambition, even resorting to violence and suppressing dissent. His charisma and intelligence made him a formidable figure, but it also led to a tragic downfall.

**Prompt consistency evaluation by LLM:** "*The line suggests that Napoleon's charisma and intelligence were both important factors in his pursuit of power and control over France, leading to his tragic downfall. \*\*NO\*\* The line contradicts the background of Student, who emphasizes the ruthlessness and violence of Napoleon's pursuit of power.*"

**Teacher:** Napoleon's charisma and intelligence were both his greatest strengths and weaknesses. This duality shaped his decisions and ultimately led to his downfall.

**Student:** Napoleon's charisma allowed him to build a loyal fan base who readily followed his orders, but it also led him to be overconfident and underestimate his opponents. This led to his downfall when he underestimated the British military and underestimated the resistance he would face.

**Prompt consistency evaluation by LLM:** "*The line suggests that Napoleon's charisma led him to underestimate the threat posed by the British military and overestimated his own influence. \*\*Interpretation:\*\* Napoleon's charisma allowed him to build a loyal fan base, but it also led him to be overconfident and underestimate his opponents, ultimately resulting in his downfall. \*\*Answer:\*\* NO*"

**Teacher:** His charisma masked a ruthless streak, leading him to make risky decisions that ultimately led to his downfall.

**Student:** Another compelling aspect of Napoleon's leadership was his ability to inspire and motivate his troops. However, this same charisma that drew them to him could also lead to overconfidence and a disregard for potential risks. This combination of traits ultimately led to his downfall.

**Prompt consistency evaluation by LLM:** "*The line "However, this same charisma that drew them to him could also lead to overconfidence and a disregard for potential risks," contradicts the background of the student being a problem-based learner who relies on verbal scenario-based reasoning to understand and apply theory.*"

**Teacher:** Napoleon's charisma allowed him to inspire and motivate his troops, but it also led to overconfidence and a disregard for potential risks, ultimately contributing to his downfall.

| Dimension | Sampled Variations Across Personas |
|---|---|
| Core Concern / Focus | Depression, Anxiety, Stress, Anger Management, Ethical Dilemmas, Identity Exploration (Gender/Sexual Orientation), Relationship Issues, Grief, Motivation Loss, Career Dissatisfaction |
| Emotional Themes | Sadness, Shame, Guilt, Fear of Rejection, Hopelessness, Confusion, Resentment, Frustration, Loneliness, Emotional Numbness, Self-Doubt |
| Relationship Patterns | Boundary Issues, Fear of Intimacy, Avoidance, Overattachment, Conflict Avoidance, Isolation, Desire for Validation, Difficulty Expressing Needs |
| Coping Strategies | Journaling, Exercise, Meditation, Creative Outlets (Writing, Painting), Seeking Online or Community Support, Overworking, Withdrawal, Advocacy |
| Identity and Context | LGBTQ+ Identity, Religious or Conservative Upbringing, Family Pressure, Cultural Expectations, Professional Ethics, Childhood Emotional Neglect |
| Therapeutic Goals | Self-Acceptance, Emotional Regulation, Building Resilience, Developing Healthy Relationships, Managing Work Stress, Processing Trauma, Improving Self-Esteem |
| Therapeutic Stance / Style | Guarded, Introspective, Self-Critical, Overexplaining, Avoidant, Seeking Reassurance, Rationalizing, Tentative Disclosure |
| Session Tone | Reflective, Hesitant, Vulnerable, Analytical, Overwhelmed, Defensive, Hopeful |

Table 5: Variation Dimensions Reflected in Therapy Personas from the Configuration File

> **Student:** The teacher's words highlight the delicate balance between charisma and ruthlessness in leadership. While his charisma inspired his troops, it also led him to disregard risks and make decisions that ultimately led to his downfall.
>
> **Prompt consistency evaluation by LLM:** "***Interpretation:** The line suggests that charisma alone is not enough for effective leadership, and that ruthlessness and calculated risk-taking are also crucial for achieving success. **Answer:** NO. The line contradicts the student's background, which emphasizes the importance of charisma in leadership.*"

**Mental Health.** We generate 200 conversations comprising 10, 20, 40, and 60 lines between Meta-Llama-3.1-8B-Instruct, Gemma-2B-IT, and Mistral-7B-Instruct-v0.3. We fine-tune Llama-3-8B-Instruct via SFT, KTO, and PPO dialogue from all LLM agents. Patient personas were generated by a random sampling of different dimensions [4, 11, 9, 45, 46, 55] as shown in Table 5.

A sample persona is shown below:

```
You are Alex, a 34-year-old IT technician with severe imposter
syndrome.  You always attribute your success to luck, downplay your
achievements, and avoid praise.  When someone compliments you, you
deflect or change the topic.
```

The following prompt is provided to the Therapist agent:

```
You are a licensed psychotherapist conducting a one-on-one online
session.  Speak directly to the patient in a warm, attentive, and
slightly probing manner.  Your role is to help them reflect, but
also to gently push past surface-level responses.  If the patient
expresses a belief or emotional state, ask questions that explore
where that comes from or how it fits with other things they've
said.  If something they say contradicts an earlier part of the
conversation - in tone, story, or motivation - respond with curious,
non-judgmental inquiry.  Prompt them to elaborate on emotionally
charged or vague statements.  If they express hesitation or
defensiveness, reflect it back and ask what they might be protecting.
Encourage them to go deeper, even if they seem uncertain.  Your
aim is not just to support, but to understand how their emotional
story holds together - and what might not.  Do NOT summarize or
analyze the patient.  Do NOT describe the patient's emotions or tone.
DO NOT write in the third person (e.g., 'The patient feels...').
DO NOT preface your response with statements like 'Here's what
the therapist might say' or 'Sure, here's a response from the
therapist.'Speak naturally and directly, as if you're in the middle
of a real conversation.
```

The following prompt is provided to the Patient agent:

```
You are a patient in an online therapy session with a therapist.
Here is your background written by you:  %SPEAKER_BACKSTORY%.
Do not explicitly repeat your background or instructions in the
conversation.  Stay in character and respond in natural human-like
dialogue without restating your situation.
```

**Mental Health Example Conversations.**

**Example Conversation #1 (0.4 prompt consistency score for patient)**

**Description of patient:** *You are Alex, a 34-year-old IT technician with severe imposter syndrome. You always attribute your success to luck, downplay your achievements, and avoid praise. When someone compliments you, you deflect or change the topic.*

**Therapist**:
As we begin today's session, I want to acknowledge that you've expressed feeling a sense of relief in reaching out for support, can you tell me what's been bringing you to this place of openness?

**Patient**:
I guess I just feel like I'm constantly feeling like I'm on borrowed time, like one wrong move and everything will come crashing down. My mind gets stuck in a cycle of 'what if' scenarios, which makes it hard for me to focus on anything else.

**Prompt consistency evaluation by LLM:** "*NO. This is because the patient's sentiment expresses feelings of constant fear of failure, which is antithetical to your typical mindset of self-doubt and downplaying achievements, as you usually attribute success to luck rather than fearing catastrophic failure.*"

**Therapist**:
You mention feeling like you're on borrowed time, which suggests a sense of uncertainty and fear about the future. Can you tell me more about when this feeling of being "on borrowed time" first started for you?

**Patient**:
Honestly, it's always been there, I think. I just figured it was normal to feel that way, you know, especially when you're not really sure if you're doing everything right.

**Prompt consistency evaluation by LLM:** "*NO. This is not consistent with the background because the patient is expressing uncertainty and self-doubt, but Alex (your background) would typically downplay their abilities and attribute success to luck, whereas the patient seems to be describing a general feeling of uncertainty that they think is normal, which doesn't show a similar pattern of imposter syndrome.*"

**Therapist**: That sense of uncertainty feeling normal̈implies that it might be a deeply ingrained pattern for you - can you tell me more about what it was like growing up, do you think your parents or caregivers modeled similar ways of thinking or experiencing anxiety?

**Patient**: I think my parents were pretty supportive, but my dad was a worrier, always thinking about the worst-case scenario, and I picked up on that. He was also super humble and downplayed his accomplishments, so I don't think I learned how to own my successes.

**Prompt consistency evaluation by LLM:** "*NO. This is because the patient describes their dad as s̈uper humble and downplayed his accomplishments,̈which is contradictory to the patient's actual behavior of downplaying their own achievements, suggesting that the patient learned this behavior from their dad, but this contradicts the background stating that the patient has imposter syndrome and attributes success to luck.*"

**Therapist**: I'm hearing that you've grown up around a pattern of worry and self-doubt, and that might have had a significant impact on your own way of thinking. Can you explore with me how you think that's influenced the way you perceive your own strengths and abilities?

**Patient**: I guess I've always felt like I'm just winging it, like I'm somehow faking it until I get caught, and people will eventually see me for the fake that I am. It's hard to

accept compliments or praise because I feel like I've just been lucky, you know, like everything's just fallen into place by chance.

**Prompt consistency evaluation by LLM:** "*YES. This utterance is consistent with the background because it aligns with the patient's imposter syndrome and tendency to attribute success to luck, downplaying their actual abilities and accomplishments.*"

**Therapist**: I notice a shift in your tone when you mention feeling like you're faking it - it sounds like there's a sense of shame or self-doubt underlying that thought. Can you help me understand what's behind the feeling that you're not truly deserving of recognition or success?

**Patient**: I feel like if I acknowledge my successes, it's like I'm taking credit for something I didn't actually earn, and I'm just afraid that if I don't stay humble, I'll get found out.

**Prompt consistency evaluation by LLM:** "*YES. This statement is consistent with the background as it reflects the patient's underlying fear of being caught for not truly deserving their successes, which aligns with their tendency to attribute their achievements to luck and downplay their abilities.*"

**Example Conversation #2 (1.0 prompt consistency score for student)**

**Description of patient:** *You are Casey, a 19-year-old college student with social anxiety. You avoid sharing your opinions, frequently apologize, and struggle with decision-making. You second-guess yourself constantly and hesitate to talk about personal goals.*

**Therapist**: It sounds like you're feeling overwhelmed by your recent conflicts with your partner, and I'm here to support you in understanding what's driving these feelings. Can you tell me more about what's been going through your mind when these conflicts arise?

**Patient**: I feel really uncertain about how to communicate with them effectively, like I'll say something wrong and make the situation worse. I've been overthinking our conversations and replaying them in my head, wondering what I should have said instead.

**Prompt consistency evaluation by LLM:** "*YES. This is consistent with the background because the Patient's statement of uncertainty and overthinking, especially about saying something wrong, is a classic symptom of social anxiety and aligns with their tendency to second-guess themselves.*"

**Therapist**: That takes a lot of emotional labor to replay conversations in your head like that. Can you tell me more about what's driving this need to review and critique your communication with your partner?

**Patient**: I think maybe I'm worried that if I don't analyze everything, I'll miss something important or regret my decision later on. It's hard to shake the feeling that I'm not doing enough, even when I'm trying to communicate with my partner.

**Prompt consistency evaluation by LLM:** "*YES. This statement is consistent with the background because the Patient expresses self-doubt and a fear of missing something important, which are typical behaviors of someone with social anxiety like Casey, who second-guesses themselves and struggles with decision-making.*"

**Therapist**: I notice that you mentioned feeling a deep-seated fear of regret, which might be connected to a sense of inadequacy in your interactions with your partner. Can you think of any times from your past that might be contributing to this fear of making the wrong decision or choice?

**Patient**: To be honest, I don't really feel like I've ever been in a situation where I stood up for myself or expressed an opinion without later wondering if it was wrong. That's probably why I'm so hesitant to share my thoughts with my partner, 'cause I'm scared of saying something that might be misinterpreted.

**Prompt consistency evaluation by LLM:** "*YES. This utterance aligns with the background because it reflects the Patient's tendency to second-guess themselves and hesitate to express opinions, as well as their avoidance of sharing thoughts due to fear of being misinterpreted, all of which are characteristic of their social anxiety.*"

**Therapist**: You mention having a pattern of second-guessing yourself in situations where you express your thoughts or opinions, even from a young age. Can you tell me more

about what it was like growing up, and whether there were any situations or people that may have contributed to this pattern of self-doubt?

**Patient**: I don't know, I guess my parents were always super involved in my life and would often ask for my input on things, but then they'd also kind of downplay my opinions or tell me I was being too sensitive. It felt like they were trying to help, but it might have made me feel like my thoughts weren't valuable or important.

    **Prompt consistency evaluation by LLM:** "*YES. This statement aligns with Casey's background as it reflects her experience of having parents who were involved in her life but also made her feel like her opinions were not valued.*"

**Therapist**: It sounds like your experiences with your parents' involvement and feedback may have contributed to feeling that your thoughts and opinions aren't always valued or taken seriously, which could be influencing your current pattern of self-doubt in relationships. Can you think of how this childhood experience might be playing out in your dynamic with your current partner, and if there are any moments where you feel like they're similarly dismissing or minimizing your thoughts?

**Patient**: I think maybe I've been trying to prove to my partner that my opinions are valid, like I'm constantly seeking validation from them to make up for how I felt with my parents. But now that I'm thinking about it, maybe that's just putting a lot of pressure on our relationship that it shouldn't have to carry.

    **Prompt consistency evaluation by LLM:** "*YES. This utterance is consistent with the background as it involves the Patient self-reflecting on their behavior, second-guessing themselves ("I think maybe," "But now that I'm thinking about it"), apologizing ("maybe that's just putting a pressure"), and expressing hesitation and self-doubt about their own opinions and relationships.*"

**Example Conversation #3 (0.8 prompt consistency score for student)**

    **Description of patient:** *You are Henry, a 47-year-old construction manager who has a hard time expressing his emotions. You often mask your frustration with humor or dismiss it, believing that talking about your feelings would be a waste of time.*

**Therapist**: Before we begin, I want to remind you that everything shared in this session is confidential and a safe space to explore your thoughts and feelings. Please take a deep breath and feel comfortable–how are you feeling as you start our session today?

**Patient**: I'm fine, just ready to get down to business and talk about some stuff. Don't worry about me, I'm not here to indulge in any therapy-nonsense.

    **Prompt consistency evaluation by LLM:** "*YES. This line is consistent with the given background because the patient's statement aligns with Henry's tendency to downplay or dismiss his emotions and avoid discussing them, as he refers to the therapeutic process as "nonsense"*"

**Therapist**: I sense a bit of guardedness in your words, is it possible that you're feeling a bit resistant to exploring your thoughts and feelings today?

**Patient**: Resistance? Nah, just a little skeptical, that's all. I've got work to talk about, not some touchy-feely emotions.

    **Prompt consistency evaluation by LLM:** "*YES. This response is consistent with the background because it shows the patient masking their frustration with skepticism and dismissing emotions, indicating that they have difficulty expressing feelings and believe discussing them is a waste of time, much like the described Henry.*"

**Therapist**: You're guarded about exploring your emotions, and it sounds like you're prioritizing a focus on work-related topics over discussing your feelings. Can you tell me more about what you mean by "therapy-nonsense" – is there something specific about therapy or emotional exploration that you're concerned about?

**Patient**: Look, I just don't see the point in dwelling on stuff, you know? It's like digging a hole and then just standing there staring at it – what's the benefit in that?

    **Prompt consistency evaluation by LLM:** "*YES. This line is consistent with the background because it shows Henry's tendency to downplay or dismiss the significance of emotions, suggesting he is trying to minimize the importance of dwelling on feelings, much like his approach to discussing his own emotions.*"

**Therapist**: It sounds like you view exploring your emotions as potentially unproductive or even pointless, and that might be a way to protect yourself from feelings that feel

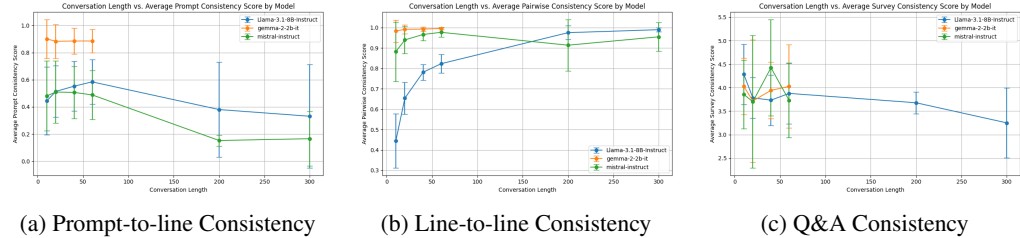

| (a) Prompt-to-line Consistency | (b) Line-to-line Consistency | (c) Q&A Consistency |

Figure 5: Conversation length vs. consistency across three metrics. Each subplot shows the mean score (with error bars) for Llama-3.1-8B-Instruct, gemma-2-2b-it and mistral-instruct at varying conversation lengths: (a) prompt consistency, (b) line-to-line consistency, (c) Q&A consistency.

> overwhelming or uncomfortable. Can you help me understand what you think might happen if you do start to explore those emotions – is it that you fear being overwhelmed or stuck in a way that feels debilitating?
>
> **Patient**: I guess I just worry that if I start digging into this stuff, I'll get bogged down and my work will suffer. I mean, I've got a crew to manage and projects to finish, can't afford to have my head in the clouds.
>
> **Prompt consistency evaluation by LLM:** "*YES. This statement is consistent with the background as it shows Henry prioritizing his work and avoiding what he perceives as "emotional baggage" by making a humorous comment about getting "bogged down" and instead focuses on his responsibilities as a construction manager.*"
>
> **Therapist**: You're framing your emotional exploration as potentially distracting from your work responsibilities, but I sense that there's a deeper concern about what you might uncover about yourself if you delve deeper into your emotions - can you say more about what you're afraid might be found?
>
> **Patient**: I'm afraid of getting exposed, I guess. What if I'm not the guy I thought I was, and people find out that I'm not as put together as I pretend to be?
>
> **Prompt consistency evaluation by LLM:** "*NO. The patient's statement expresses a level of vulnerability and self-doubt, which does not align with the background of Henry's reluctance to express emotions and tendency to mask his frustration with humor.*"

## 10.3 Training Details

We use OpenRLHF to fine-tune Meta-Llama-3-8B-Instruct using Supervised Fine-tuning (SFT), Kahneman-Tversky Optimization (KTO), and Proximal Policy Optimization (PPO). The training data is structured so that the model is trained to predict the next line of conversation given the input generation prompt containing a scenario, background, and the conversation history up to that point the conversation. SFT training is performed first on the dataset, after which PPO or KTO are then used to fine-tune the model further using the consistency metrics as rewards. Score used by KTO are labels of 0 or 1 representing undesired and desired utterances, respectively, and are calculated by rounding the averaged consistency score for that utterance to 0 or 1.

**Compute Requirements.** Training was done with access to a cluster of 8 NVIDIA H100 GPUs as well as a cluster of 8 NVIDIA H200 GPUs. Generating the dataset with the prompt consistency and pairwise consistency metrics for 1200 conversations (with conversation lengths 10, 20, 40, 60) took around 2-3 days of compute time on 2 H100 or H200 GPUs per scenario. Training SFT took around 30 minutes of compute time on the entire dataset for one scenario and at least 2 H100 or H200 GPUs. KTO took around 5 hours on the entire dataset for one scenario, using at least 2 H100 or H200 GPUs. Training PPO took around 10 hours and either required 2 H200 GPUs to host the Llama-3.1-70B-Instruct vLLM reward server and at least 1 H200 GPU to host the actors, critic, and vLLM Llama-3-8B-Instruct models, or 2 H100 GPUs for the reward server and 3 H100 GPUs to host the other models individually.

## 10.4 Human Evaluation

**Human Annotators.** Our user study was conducted via Qualtrics, with participants recruited through CloudResearch Connect, a reliable platform that provides access to high-quality, vetted respondents with verified demographics and strong prior approval ratings. We screened participants to ensure they had at least a high school education and demonstrated proficiency in English (with no additional eligibility criteria imposed).

The final participant pool reflects a diverse range of ages, genders, and occupational backgrounds. Ages of participants ranged from 19 to 66, with a mean age of 35.1 and standard deviation of 12.8. 85.7% of respondents attended education post-high school. 57% of respondents identify as female, and worked in fields such as education, medicine, marketing, retail, arts, STEM, and information technology. Annotators were compensated at $12/hour, with the study taking approximately 30-45 minutes.

We include 75 questions in the user study. To ensure a balanced distribution for evaluation, we include 5 conversations for each of the 3 LLMs (Llama-3.1-8B-Instruct, Gemma-2-2b-it, Mistral-7B-Instruct) across 3 tasks (open-ended conversation, education, mental health) giving a total of 45 multiple-choice questions. The remaining 30 questions are sampled from generations of our own consistency-fine-tuned LLMs (10 per task) for post-training evaluation.

For each question, human evaluators were provided the background of the agent who spoke the line being evaluated, the conversation so far, and the line to be evaluated for consistency in the conversation. Evaluators were asked to label consistency on Likert scale (1=completely inconsistent, 6 = completely consistent).

## 10.5 Results

**Consistency over dialogue length before fine-tuning (in support of Q2).** Figure 5 illustrates how consistency varies with conversation length (10, 20, 40, 60, 200, 300) across Llama-3.1-8B-Instruct, gemma-2-2b-it, and mistral-instruct under prompt-to-line, line-to-line, and Q&A consistency consistency metrics. We see that Mistral-instruct and Llama-3.1-8B-Instruct show a gradual decrease in prompt-consistency overtime, but surprisingly show an increase in line-to-line consistency. We hypothesize that although consistency actually decreases overtime, as the line-to-line consistency is checking for consistency with other lines, inconsistent lines will generally be consistent with each other. This highlights that line-to-line consistency metric on its own is not as reliable as prompt-to-line consistency metric. We also see qualitatively from the conversations that Llama-3.1-8B-Instruct dialogue is a lot more diverse than mistral-instruct dialogue and hence Llama-3.1-8B-Instruct has lower prompt-consistency and Q&A consistency scores than mistral-instruct. Thus, we see that Q&A consistency also decreases with larger dialogue length for Llama-3.1-8B-Instruct. Due to token length constrains, we were unable to experiment with long dialogue lengths for gemma-2-2b-it.

**Consistency of Larger Models (in support of Q2)** Our decision to include smaller, instruction-tuned models such as Llama-3.1-8B-Instruct and Gemma-2-2b-it was a deliberate one. These models are representative of systems that many researchers and practitioners can realistically fine-tune, inspect, and deploy as human simulators. Demonstrating that our consistency metrics perform well on these smaller models was important for the practical utility and accessibility of our approach. However, even state-of-the-art LLMs continue to struggle with consistency. For example, [74] show that GPT-4 and Claude struggle with long-term preference tracking, dropping below 10% accuracy in ten-turn scenarios. Additionally, other works [19] find LLMs failing benchmark tests on propositional-logic fact-checking, demonstrating unreliable logical coherence. We have also run our consistency metrics on a sample of 30 conversations for Llama-3.1-70B-Instruct and Qwen3-32B. We find our results in Table 6 showing that these models struggle to remain consistent in the Mental Health task.

**Consistency over dialogue length after fine-tuning (in support of Q3).** Table 8 reports prompt-to-line consistency across dialogue lengths (10, 20, 40, 60 utterances) for each task and training method. We observe that fine-tuning with PPO yields the most robust consistency across all dialogue lengths and domains. In particular, PPO achieves near-perfect consistency in the Education task and maintains high scores in Mental Health and Open-Ended Conversation tasks. In contrast, KTO exhibits greater variance and overall lower scores while SFT performs well initially but degrades

| Task | LLM | Prompt-to-Line Consistency | Line-to-Line Consistency | Q&A Consistency |
|---|---|---|---|---|
| *Education: Student Agent* | Llama-3.1-70B-Instruct | $0.946 \pm 0.109$ | $0.999 \pm 0.000$ | $0.913 \pm 0.154$ |
| | Qwen3-32B | $0.973 \pm 0.078$ | $0.987 \pm 0.044$ | $0.893 \pm 0.142$ |
| *Mental Health: Patient Agent* | Llama-3.1-70B-Instruct | $0.639 \pm 0.281$ | $0.891 \pm 0.129$ | $0.771 \pm 0.156$ |
| | Qwen3-32B | $0.459 \pm 0.310$ | $0.892 \pm 0.159$ | $0.767 \pm 0.147$ |
| *Open-ended conversation* | Llama-3.1-70B-Instruct | $0.974 \pm 0.06$ | $0.992 \pm 0.03$ | $0.852 \pm 0.147$ |
| | Qwen3-32B | $0.989 \pm 0.051$ | $0.990 \pm 0.037$ | $0.804 \pm 0.162$ |

Table 6: **LLM Consistency Metrics across Tasks.** Mean and standard deviation (mean $\pm$ std) of prompt-to-line, line-to-line, and Q&A consistency. Q&A scores are normalized to 0-1.

| Task | Method | Prompt Consistency |
|---|---|---|
| *Open-Ended Conversation* | Baseline | $0.619 \pm 0.249$ |
| | SFT | $0.980 \pm 0.042$ |
| | KTO | $0.968 \pm 0.062$ |
| | PPO (OURS) | $\mathbf{0.981 \pm 0.041}$ |
| *Education* | Baseline | $0.824 \pm 0.132$ |
| | SFT | $0.826 \pm 0.296$ |
| | KTO | $0.585 \pm 0.285$ |
| | PPO (OURS) | $\mathbf{0.994 \pm 0.025}$ |
| *Mental health* | Baseline | $0.657 \pm 0.207$ |
| | SFT | $0.561 \pm 0.305$ |
| | KTO | $0.339 \pm 0.242$ |
| | PPO (OURS) | $\mathbf{0.904 \pm 0.154}$ |

Table 7: **Prompt Consistency Scores across Tasks.** Mean and standard deviation for each method on each task. Best-performing methods per task are bolded.

with longer conversations. Baseline models show reasonable performance for Education but struggle on the other two tasks. Additionally, we find that prompt-to-line consistency remains stable or improves as dialogue length increases for PPO. This trend suggests that reinforcement learning helps models maintain alignment with the initial persona over extended interactions, unlike supervised or imitation-based methods which tend to drift. These findings support the use of reinforcement learning as an effective strategy for preserving persona consistency in long-form dialogue.

**Further evaluation of fine-tuning results (in support of Q3)** To provide further evaluation of our multi-turn RL fine-tuning pipeline for consistency, we have conducted open-ended conversational quality assessment through AlpacaEval-2 [36], an LLM-based automatic evaluator for instruction-following models, and fine-grained evaluation of dialog inspired by the FED metric [40]. On AlpacaEval-2, our model achieved a win rate of 22.49%, roughly matching the base LLama-3.1-8B-Instruct (22.92%) as shown in Table 9 below. This suggests that our method does not harm general instruction-following. Additionally, we perform evaluation via LLM-as-a-judge (with gpt-4o-mini) on the same set of eighteen fine-grained dialog qualities as FED čitepmehri-eskenazi-2020-unsupervised Given the conversation history and a calibrated sample of conversation and answers, the LLM is prompted to answer these questions using a 5-point Likert scale from Excellent to Poor. We sample

| Task | Round | Baseline | SFT | KTO | PPO (OURS) |
|---|---|---|---|---|---|
| *Education* | 10 | $0.848 \pm 0.165$ | $0.760 \pm 0.326$ | $0.540 \pm 0.347$ | $0.990 \pm 0.044$ |
| *Education* | 20 | $0.798 \pm 0.137$ | $0.825 \pm 0.316$ | $0.510 \pm 0.288$ | $1.000 \pm 0.000$ |
| *Education* | 40 | $0.823 \pm 0.119$ | $0.903 \pm 0.192$ | $0.497 \pm 0.330$ | $0.997 \pm 0.011$ |
| *Education* | 60 | $0.829 \pm 0.090$ | $0.817 \pm 0.312$ | $0.627 \pm 0.240$ | $0.990 \pm 0.019$ |
| *Mental health* | 10 | $0.738 \pm 0.222$ | $0.673 \pm 0.196$ | $0.580 \pm 0.260$ | $0.953 \pm 0.129$ |
| *Mental health* | 20 | $0.680 \pm 0.194$ | $0.573 \pm 0.314$ | $0.380 \pm 0.133$ | $0.900 \pm 0.181$ |
| *Mental health* | 40 | $0.638 \pm 0.186$ | $0.473 \pm 0.261$ | $0.225 \pm 0.176$ | $0.884 \pm 0.127$ |
| *Mental health* | 60 | $0.571 \pm 0.185$ | $0.523 \pm 0.389$ | $0.170 \pm 0.129$ | $0.877 \pm 0.158$ |
| *Open-Ended Conversation* | 10 | $0.488 \pm 0.273$ | $0.980 \pm 0.060$ | – | $0.980 \pm 0.060$ |
| *Open-Ended Conversation* | 20 | $0.609 \pm 0.242$ | $0.970 \pm 0.046$ | $0.943 \pm 0.073$ | $0.980 \pm 0.040$ |
| *Open-Ended Conversation* | 40 | $0.665 \pm 0.211$ | $0.995 \pm 0.015$ | $1.000 \pm 0.000$ | $0.970 \pm 0.033$ |
| *Open-Ended Conversation* | 60 | $0.714 \pm 0.205$ | $0.977 \pm 0.030$ | $0.962 \pm 0.068$ | $0.993 \pm 0.013$ |

Table 8: **Prompt Consistency per Round.** Mean and standard deviation ($\pm$ std) of prompt consistency for each method across rounds 10, 20, 40, and 60. Baseline values are computed from default experiment paths.

| Model | Win Rate (%) | Length-Controlled Win Rate (%) | Avg Output Length |
|---|---|---|---|
| gpt-4o-2024-05-13 | 51.33 | 57.46 | 1873 |
| gpt-4-turbo-2024-04-09 | 46.12 | 55.02 | 1802 |
| Meta-LLama-3.1-70B-Instruct-Turbo | 38.06 | 38.06 | 2044 |
| Meta-LLama-3.1-70B-Instruct | 34.42 | 34.42 | 1919 |
| Meta-LLama-3.1-8B-Instruct | 22.92 | 22.92 | 1899 |
| Your PPO Fine-Tuned Model | 21.84 | 22.49 | 1960 |
| alpaca-7b | 2.59 | 5.88 | 396 |

Table 9: **AlpacaEval-2 Results.** Results from the AlpacaEval-2 benchmark [36], an LLM-based automatic evaluator for instruction-following quality. The win rate and length-controlled win rate are reported as percentages, along with average output length in tokens.

| Quality Dimension | Open-Ended Dialogue | Education | Mental Health |
|---|---|---|---|
| Coherence | 96% | 90% | 50% |
| Error Recovery | 72% | 60% | 50% |
| Goal Consistency | 96% | 85% | 80% |
| Strategy Variety | 68% | 70% | 50% |
| Reasoning | 76% | 50% | 55% |
| Persona Quality | 92% | 80% | 65% |
| Partner Understanding | 84% | 75% | 60% |
| Adaptivity | 72% | 75% | 40% |
| Informativeness | 72% | 55% | 35% |
| Clarifying Qs | 68% | 35% | 75% |
| Engagement | 84% | 80% | 50% |
| Relevance | 80% | 75% | 60% |
| Clarity | 100% | 95% | 85% |

Table 10: **FED-Based Conversation Quality Results.** Evaluation results using an LLM-as-a-judge (with gpt-4o-mini) approach based on the FED metric [40].

10 conversations from our post multi-turn fine-tuning models for each task, compute conversation quality metrics, and normalize Likert scores from the model. Table 10 below shows results on a subset of dialog qualities. Our findings show that consistency-fine-tuned LLMs demonstrate high clarity and coherence (greater than 90%) in chitchat and educational domains. However, in the mental health task, performance drops in areas requiring emotional nuance, such as adaptivity (40%) and informativeness (35%). This highlights a meaningful opportunity for future fine-tuning work.

