# OpenReview forum: "Consistently Simulating Human Personas with Multi-Turn Reinforcement Learning"
_NeurIPS.cc/2025/Conference — NeurIPS 2025 poster_

### Official Review · Reviewer_qLt6 · 2025-06-18

**Clarity:** 4
**Significance:** 3
**Originality:** 3
**Rating:** 5
**Confidence:** 4

**Summary:**

The authors propose 3 consistency metrics to measure how coherently an LLM represents a human person. Then authors then use an LLM-as-a-judge implementation of these metrics as a reward signal to finetune 3 light models with PPO, and show the inconsistency reduced roughly twice. The authors study the alignment of LLM-as-a-Judge decisions with human experts/annotators.

**Questions:**

[1] In Table 1 I presume, there are metrics “after”, i.e. finetuned. Where do I find the baseline “unfinetuned” metrics?

[2] What model was used for LLM-as-a-Judge?

[3] How long did the RL finetuning take and on what hardware?

[4] What was the bottleneck in the training: episode generation, LLM-as-a-Judge reward generation, or the optimizer step?

Comments:

[1] In Figure 4 “(PPO, ours)” would look better as “PPO, chosen by us”

[2] “Li et al 2023, CAMEL: Communicative Agents for "Mind" Exploration of Large Language Model Society” should be cited as it introduces the role-playing setup with LLMs and notes that LLMs often swerve from following the assigned role.

[3] “Prior work on the working memory capacity of LLMs shows that while these models demonstrate strong short-term recall, they struggle with maintaining information over longer contexts, indicating limited long-term memory” the source of low long term memory could be discussed, for example “Kuratov et al 2024. BABILong: Testing the Limits of LLMs with Long Context Reasoning-in-a-Haystack”.

**Ethical Concerns:**

["NO or VERY MINOR ethics concerns only"]

**Final Justification:**

The authors have addressed my concerns.

**Limitations:**

[1] Given the recent advances in frontier LLMs, the need for RL FT of models for better consistency may go away. The frontier LLMs can be finetuned as well, but the increase in consistency may be negligible for them.

**Quality:**

3

**Strengths And Weaknesses:**

Strengths:

[1] The paper brings up the topic of simulating/mocking the users, which has a lot of practical value for testing/evaluating LLM-based products.

[2] The paper is clearly written, with a clear discussion of the problem, the proposed solution and the experiments.

[3] Figure 2 demonstrates the problem at hand well.

[4] The very idea to finetune a model for a niche purpose with RL is intriguing and the fact that it succeeds (at least for small, very far from frontier, models) is inspiring.

Weaknesses:

[1] Preserving consistency excludes the possibility of some complex scenarios, specifically, a person's intention to lie and deceive, which may be a common occurrence in psychology, customer support, and education.

[2] The models studied are small (2-8B) and old. ​​Llama-8B-Instruct is the first generation Llama, whereas the last adequate generation is the third. There is a high chance that the recent better trained models of the same size, or bigger (14-32b) models match or surpass the finetuned models in consistency by the sheer cognitive power and precise instruction following. These experiments are necessary. Using small LMs must be justified, as there are plentiful large (70B+) LMs out there.

---

> ### Author Rebuttal · Authors · 2025-07-31
>
> We thank you for your constructive and thorough feedback, as well as comments regarding prior work and formatting. We will be sure to include these in our revised paper. We have addressed the main concerns raised in your review by (1) discussing how to handle complex scenarios that may include lying and deception from the simulated human (2) clarifying our use of smaller models and providing new results with Llama-3.1-70B-Instruct (3) answering questions you had on our LLM-as-a-Judge framework and compute requirements. We have also performed a larger user study with 30 annotators, as discussed in response #1 to reviewer GnPa.
>
> 1) *"[1] Preserving consistency excludes the possibility of some complex scenarios, specifically, a person's intention to lie and deceive, which may be a common occurrence in psychology, customer support, and education.”*
>
> We thank you for raising this important point. We agree that human communication can involve intentional deception, omission, or strategic inconsistency, particularly in domains like psychology or customer support. Our framework is fully compatible with simulating such behaviors when they are part of the intended persona. For example, if the goal is to simulate an agent that lies or withholds information, this can be explicitly encoded into the persona prompt (e.g., "You often conceal the truth" or "You occasionally give misleading answers"). In this case, our consistency metrics will still function as intended, measuring internal coherence with that defined behavior, rather than truthfulness. This is illustrated in our therapy task, where some personas are instructed to “avoid discussing emotional issues”. Human annotators and our LLM-as-a-Judge annotator would hence mark statements that suddenly reveal vulnerability as inconsistent, not because vulnerability is bad, but because it would contradict the stated persona.
>
> 2) *"[2] The models studied are small (2-8B) and old. ​​Llama-8B-Instruct is the first generation Llama, whereas the last adequate generation is the third. There is a high chance that the recent better trained models of the same size, or bigger (14-32b) models match or surpass the fine-tuned models in consistency by the sheer cognitive power and precise instruction following. These experiments are necessary. Using small LMs must be justified, as there are plentiful large (70B+) LMs out there.”*
>
> We acknowledge your concern regarding model scale. We would like to note that our decision to include smaller, instruction-tuned models such as Llama-3.1-8B-Instruct and Gemma-2-2b-it was a deliberate one. These models are representative of systems that many researchers and practitioners can realistically fine-tune, inspect, and deploy as human simulators. Demonstrating that our consistency metrics perform well on these smaller models was important for the practical utility and accessibility of our approach. However, even state-of-the-art LLMs continue to struggle with consistency. For example, Zhao et al. \[1] show that GPT-4 and Claude struggle with long-term preference tracking, dropping below 10% accuracy in ten-turn scenarios. Additionally, other works \[2] find LLMs failing benchmark tests on propositional-logic fact-checking, demonstrating unreliable logical coherence.As per your comment, we have also run our consistency metrics on a sample of 30 conversations for Llama-3.1-70B-Instruct and Qwen3-32B. We find our results below, showing that these models struggle to remain consistent in the therapy task. We will report these results in the revision.
>
> | Task                         | Model                  | Prompt-to-Line Consistency | Line-to-Line Consistency | Q&A Consistency |
> |------------------------------|------------------------|----------------------------|--------------------------|-----------------|
> | Education: Student Agent     | Llama-3.1-70B-Instruct | 0.946 ± 0.109              | 0.999 ± 0.000            | 0.913 ± 0.154   |
> | Education: Student Agent     | Qwen3-32B              | 0.973 ± 0.078              | 0.987 ± 0.044            | 0.893 ± 0.142   |
> | Mental Health: Patient Agent | Llama-3.1-70B-Instruct | 0.639 ± 0.281              | 0.891 ± 0.129            | 0.771 ± 0.156   |
> | Mental Health: Patient Agent | Qwen3-32B              | 0.459 ± 0.310              | 0.892 ± 0.159            | 0.767 ± 0.147   |
> | Open-ended Conversation      | Llama-3.1-70B-Instruct | 0.974 ± 0.060              | 0.992 ± 0.030            | 0.852 ± 0.147   |
> | Open-ended Conversation      | Qwen3-32B              | 0.989 ± 0.051              | 0.990 ± 0.037            | 0.804 ± 0.162   |
>
> References:
>
> [1] Evaluating Personalized Preference Following in LLMs (2025). <https://arxiv.org/abs/2502.09597>
>
> [2] Logical Consistency of Large Language Models in Fact-Checking (2024). <https://arxiv.org/abs/2412.16100>
>
> 3) *“\[1] In Table 1 I presume, there are metrics “after”, i.e. finetuned. Where do I find the baseline “unfinetuned” metrics?”*
>
> We would like to clarify that Table 1 reports results for popular, widely deployed instruction-tuned models (Llama-3.1-8B-Instruct, Gemma-2-2B-It, and Mistral-7B-Instruct-v0.3) that were not fine-tuned by our method for consistency. Results after applying our consistency-driven fine-tuning using multi-turn RL are presented in Figure 4. We treat Llama-3.1-8B-Instruct as the primary baseline, as it represents a strong, openly available instruction-tuned model. We intentionally exclude non-instruct-tuned models like Llama-3.1-8B from our evaluation, as they fail to generate coherent and high-quality conversations across both general and task-specific settings when compared to instruction-tuned LLMs.
>
> 4) *“\[2] What model was used for LLM-as-a-Judge?”*
>
> We use Llama-3.1-70B-Instruct as our consistency evaluator (LLM-as-a-Judge). While this is noted in Appendix Section A, we will move this detail to the main Experiments section in the revised paper for improved clarity.
>
> 5) *“\[3] How long did the RL finetuning take and on what hardware?”*
>
> As noted in the Appendix (under Section 8.3, Compute Requirements), training was done with access to a cluster of 8 NVIDIA H100 GPUs as well as a cluster of 8 NVIDIA H200 GPUs. The training data is structured so that the model is trained to predict the next line of conversation given the input generation prompt containing a scenario, background, and the conversation history up to that point in the conversation. SFT training is performed first on the dataset, after which PPO or KTO are then used to fine-tune the model further using the consistency metrics as rewards. For a rough estimate of time, the process to train a PPO model, from generating training data to training time, took 2-3 days for each task.
>
> 6. *“[4] What was the bottleneck in the training: episode generation, LLM-as-a-Judge reward generation, or the optimizer step?”*
>
> The primary bottleneck in our training pipeline was LLM-as-a-Judge reward generation, as computing consistency scores required heavy querying of Llama-3.1-70B-Instruct after every step in the conversation generation process.

---

> > ### Comment · Reviewer_qLt6 · 2025-08-06
> >
> > Dear authors, thank you for addressing my concerns. I have raised my score.

---

> ### Author Response · Authors · 2025-08-05
>
> Thank you for taking the time to read and acknowledge our rebuttal. We would appreciate if you could let us know whether our response addressed your concerns! If there are any remaining questions, we would be happy to provide further clarification.

---

### Official Review · Reviewer_GnPa · 2025-06-30

**Clarity:** 3
**Significance:** 2
**Originality:** 3
**Rating:** 4
**Confidence:** 4

**Summary:**

This paper introduces a framework to evaluate and improve the persona consistency of Large Language Models (LLMs) used as human simulators. The authors propose three automated metrics—prompt-to-line, line-to-line, and Q&A consistency—which are evaluated by a separate LLM-as-a-Judge. These metrics are then used as reward signals in a multi-turn reinforcement learning (RL) process (using PPO) to fine-tune simulator agents to be more consistent. The authors test their approach in three domains (open-ended conversation, education, and mental health) and show that their method reduces inconsistencies compared to baseline models and other fine-tuning techniques.

**Questions:**

Undisclosed Judge Model: Could you please specify the proprietary model used as the LLM-as-a-Judge? Given that this component is central to your entire framework, its identity is crucial for reproducibility and for the community to understand the potential biases of the reward signal. Why was this information omitted?

The Judge Paradox in Mental Health: In the Mental Health domain, your automated metric showed no meaningful correlation with human judgments. Yet, the model fine-tuned with PPO using this metric still showed a large improvement (Figure 4). How do you explain this?

Human Annotator Curation: Could you provide details on the human annotators for your validation study? What were their qualifications? How were they sourced and compensated? This information is essential for assessing the quality and reliability of your human baseline, which you use to validate your core metrics.

Choice of Base Models: Your experiments use older and relatively weak models. Given that state-of-the-art proprietary models are accessible via APIs (often with free tiers/credits for research), could you justify the decision not to include at least one stronger model as a baseline? This would significantly strengthen the claim that your findings are relevant to current systems.

Evolving Personas: Your framework penalizes all deviations from an initial, static persona. Have you considered how to adapt this to distinguish between undesirable contradictions and plausible character evolution?

**Ethical Concerns:**

["NO or VERY MINOR ethics concerns only"]

**Final Justification:**

The authors provided clear clarifications and meaningful additions to address my concerns about transparency and the definition of consistency. They moved key experimental details to the main text for clarity, benchmarked their LLM-as-a-Judge approach against human annotators with strong alignment results, and openly acknowledged the current limitations of their framework. Their planned extensions toward modeling dynamic, context-sensitive behavior significantly strengthen the potential impact of this work. These improvements justify increasing my score.

**Limitations:**

Use of Outdated Models and a Flawed Human Baseline: The paper's experimental foundation is weak. The work relies on small and outdated models (e.g., Llama-8B, Gemma-2B) whose behaviors are not representative of the state-of-the-art. The argument for their use (cost) is unconvincing, as more powerful proprietary models are readily accessible via APIs, which often include generous free tiers and research credits. Furthermore, the human validation, which is meant to be the ground truth, is critically insufficient. It is based on a tiny sample of 30 dialogues and just three annotators. The paper provides no details on who these annotators were, their level of expertise, or how they were sourced and instructed. This lack of transparency makes it impossible to assess the quality of the human baseline, which may well be based on low-quality work from unvetted crowd-workers.

The Unexplained Judge Paradox: The paper presents a paradoxical finding without any attempt at an explanation. In the Mental Health domain, the authors report that the initial LLM-as-a-Judge's evaluations had a weak and negative correlation (r = -0.035) with human judgments. This means the judge was, by their own measure, unreliable. However, after using this unreliable judge as the reward signal for PPO fine-tuning, the resulting model’s dialogues suddenly achieved a strong positive correlation (r = 0.476) with human judgments. The paper fails to discuss how a demonstrably flawed reward signal can lead to a verifiably better outcome. This is a major analytical gap. It raises questions as to whether the model simply learned to "game" the flawed metric or if the result is a statistical artifact of the tiny human evaluation set. This critical point needs to be addressed.

A Rigid and Unrealistic Definition of Consistency: The framework defines consistency as a rigid, robotic adherence to a static persona. This is a significant limitation, as real human interaction involves nuance, growth, and justified changes in perspective. The paper should acknowledge that it is optimizing for a narrow form of consistency that may not translate to realistic and engaging human simulation, and it does not account for plausible character evolution.

**Quality:**

2

**Strengths And Weaknesses:**

Strengths:

*   Significance: The paper addresses a critical and timely problem. As LLMs are increasingly used as proxies for humans, ensuring their behavioral consistency is vital for the reliability of downstream applications, from AI agent training to social science research. The direction of using automated pipelines to improve and benchmark this capability is highly valuable.
*   Originality: The core methodological contribution—using multi-turn RL with turn-level rewards derived from an LLM-as-a-Judge to explicitly optimize for dialogue consistency—is a novel and well-conceived approach. It provides a scalable alternative to manual data annotation and prompt engineering.
*   Clarity: The paper is well-written, and the proposed framework, including the three consistency metrics and the RL setup, is explained clearly and logically. The figures, particularly Figure 2, provide excellent, intuitive examples of the types of inconsistencies the metrics are designed to catch.

Weaknesses:

*   Weak Models: The experiments rely on relatively small and older instruction-tuned models (Llama-8B, Gemma-2B, etc.) that are not representative of the current state of the art. While the authors argue this is for cost reasons, conclusions drawn from these models may not generalize to more capable systems. The feasibility of using stronger models, even through APIs with free research credits, is not adequately explored.

*   Lack of Transparency: The paper does not disclose the specific proprietary model used for the crucial LLM-as-a-Judge component. This is a major omission that hinders reproducibility and makes it impossible for the community to fully assess the validity and potential biases of the reward signal, which is the cornerstone of the entire method.
*   Critically Flawed Human Evaluation: The human validation of the metrics is extremely limited (30 dialogues, 3 annotators). This sample is too small to draw robust conclusions. Furthermore, the paper provides no details on the sourcing, expertise, or curation of the annotators. Without this information, the quality of the "human baseline" is questionable, and it's impossible to know if it suffers from low-quality annotations from unvetted crowd-workers (e.g., from MTurk).
*   Unreliable Reward Signal: The paper's own results show a near-zero, negative correlation between the automated metric and human judgments in the sensitive Mental Health domain (r = -0.035). This is a critical failure, as it means the model was optimized against a meaningless (or even harmful) reward signal in this context. It severely undermines the paper's claims about the framework's utility in sensitive domains.
*   Rigid Definition of Consistency: The framework operationalizes consistency as a rigid adherence to a static persona. This is a "robotic" view that does not account for natural, justified evolution in a person's beliefs or tone during a conversation.

---

> ### Author Rebuttal · Authors · 2025-07-31
>
> We appreciate your critical feedback and recognition of our paper addressing a timely and important problem. We have addressed the main concerns raised in your review by (1) expanding the human evaluation to 30 annotators and reporting both inter-rater reliability and alignment with LLM judgments, (2) clarifying our use of smaller models and providing a new evaluation of Llama-3.1-70B-Instruct, (3) providing additional details of our LLM-as-a-Judge setup, and (4) explicitly acknowledging the limitations of static persona definitions, which we now include in the paper’s Limitations section.
>
> 1) *“The human validation of the metrics is extremely limited (30 dialogues, 3 annotators) \[...] details on the sourcing, expertise \[...] of the annotators.”*, *“Unreliable Reward Signal: The paper's own results show a near-zero, negative correlation between the automated metric and human judgments in the sensitive Mental Health domain (r = -0.035)..."*
>
> Following your suggestion, we have expanded our user study sample to include 30 annotators. We present our findings as follows:
>
> **User Study Description:** Our human evaluation of dialogue consistency consists of three parts: (1) a free-form question asking annotators to define what constitutes consistency in dialogue, (2) a rating task where annotators assess 15 statements in terms of importance to maintaining consistency in conversation, (3) 75 multiple-choice questions where annotators rate LLM-generated dialogue snippets from conversations on a Likert scale (1 = completely inconsistent, 6 = completely consistent). To ensure a balanced distribution for evaluation, we include 5 conversations for each of the 3 LLMs (Llama-3.1-8B-Instruct, Gemma-2-2b-it, Mistral-7B-Instruct) across 3 tasks (open-ended conversation, education, mental health) giving a total of 45 multiple-choice questions. The remaining 30 questions are sampled from generations of our own consistency-fine-tuned LLMs (10 per task) for post-training evaluation.
>
> **Human Annotators.** Our user study was conducted via Qualtrics, with participants recruited through CloudResearch Connect, a reliable platform that provides access to high-quality, vetted respondents with verified demographics and strong prior approval ratings. We screened participants to ensure they had at least a high school education and demonstrated proficiency in English (with no additional eligibility criteria imposed). The final participant pool reflects a diverse range of ages, genders, and occupational backgrounds. Annotators were compensated at $12/hour, with the study taking approximately 30-45 minutes. As per your feedback, we will report detailed annotator statistics in the Appendix of the final version of the paper.
>
> **Evaluation Procedure.** To assess the alignment between human annotators and LLM annotation, we convert Likert-scale ratings (1–6) into binary outputs representing consistent (ratings ≥ 4) and inconsistent (ratings ≤ 3) responses. We computed % agreement to capture the raw alignment between LLM predictions and human judgments across three consistency metrics: prompt consistency, line-to-line consistency, and Q\&A consistency. We report Fleiss’ kappa, widely used to assess inter-rater reliability among multiple annotators for categorical judgments \[1,2], for both human-to-human and LLM-to-human agreement. This allows us to evaluate whether the LLM aligns with human consensus at a level comparable to or exceeding that of the human annotators themselves.
>
> **Results.** From the table below, we find that our LLM annotator (LLaMA-70B-Instruct) achieves an average Fleiss’ kappa of 0.400 (± 0.092) across tasks, which exceeds human–human Fleiss’ kappa (average of 0.063 (± 0.049)) in all cases. Similarly, LLM-annotator percent agreement averaged 76.73%, which is higher than the average pairwise agreement of 69.16% between human annotators. Regarding the mental health results in Experiment 3, we agree that the original correlation between the metric and human judgments was not statistically significant. In this revised version, we expand the study with more conversations and a larger annotator pool, and find a Fleiss’ kappa of 0.453 (± 0.0078) and pairwise agreement of 88.18% for prompt-consistency, comparing reasonably well with human inter-rater agreement, with a low Fleiss’ kappa of 0.259 (± 0.062) and pairwise agreement of 74.93%. Therefore, we believe the problem results from human annotators' inability to parse inconsistency in this domain. This user study also supports our decision to choose to perform multi-turn RL fine-tuning with the prompt consistency metric, as humans most agree with this definition of consistency. Compared to line-to-line and Q\&A consistency, this metric is the least costly to compute.
>
> **Table 1: Human-LLM Agreement**
> | Task          | Consistency Definition   | % Agreement (Model–Human) | Fleiss’ Kappa (± SD) | p-value  |
> |---------------|--------------------------|---------------------------|----------------------|----------|
> | Chit-Chat     | Prompt Consistency       | 74.55%                    | 0.504 ± 0.080        | 2.97e-10 |
> | Chit-Chat     | Line-to-Line Consistency | 94.32%                    | 0.470 ± 0.083        | 1.35e-08 |
> | Chit-Chat     | Q&A Consistency          | 49.60%                    | 0.504 ± 0.069        | < 0.001  |
> | Education     | Prompt Consistency       | 90.00%                    | 0.697 ± 0.084        | 0.000    |
> | Education     | Line-to-Line Consistency | 89.77%                    | 0.713 ± 0.083        | 0.000    |
> | Education     | Q&A Consistency          | 71.86%                    | 0.459 ± 0.074        | < 0.001  |
> | Mental Health | Prompt Consistency       | 88.18%                    | 0.453 ± 0.078        | 6.21e-09 |
> | Mental Health | Line-to-Line Consistency | 89.77%                    | 0.671 ± 0.067        | 0.000    |
> | Mental Health | Q&A Consistency          | 78.18%                    | 0.444 ± 0.065        | < 0.001  |
> | **Average**   | **All Types**            | **80.03%**                | **0.552 ± 0.076**    | —        |
>
> **Table 2: Human-Human Agreement (inter-rater reliability)**
> | Task            | Pairwise Agreement (Humans) | Fleiss’ Kappa (± SE) | p-value  |
> |-----------------|-----------------------------|----------------------|----------|
> | Chit-Chat       | 71.34%                      | 0.213 ± 0.052        | 4.44e-05 |
> | Education       | 70.82%                      | 0.242 ± 0.057        | 2.26e-05 |
> | Mental Health   | 74.93%                      | 0.259 ± 0.062        | 2.67e-05 |
> | Q&A Consistency | 74.55%                      | 0.497 ± 0.083        | 2.20e-09 |
> | **Average**     | **72.91%**                  | **0.303 ± 0.064**    | —        |
>
> References:
>
> [1] Fleiss, J. L. (1971). _Measuring nominal scale agreement among many raters_. Psychological Bulletin, 76(5), 378–382. https\://doi.org/10.1037/h0031619\
> [2] Landis, J. R., & Koch, G. G. (1977). _The measurement of observer agreement for categorical data_. Biometrics, 33(1), 159–174. https\://doi.org/10.2307/2529310
>
> Due to space limitations, we will include additional details about our user study in the Appendix for reference.
>
> 2) *“Weak Models: The experiments rely on relatively small and older instruction-tuned models \[...]”*
>
> We acknowledge your concern regarding model scale and conducted new experiments with larger models (32B, 70B). We respond to your comment in reviewer qLt6 #2 response.
>
> 3) *“Lack of Transparency: The paper does not disclose the specific proprietary model used for the crucial LLM-as-a-Judge component.”*
>
> We appreciate your concern regarding the transparency of our LLM-as-a-Judge setup. We use Llama-3.1-70B-Instruct as our consistency evaluator. While this is noted in Appendix Section A, we will move this detail to the main Experiments section in the revised paper for improved clarity, based on your feedback. As with any LLM-based evaluation, some bias may be inherited from the underlying model. To mitigate this, we benchmark its judgments against human annotators as shown in #1. Our results show that Llama-3.1-70B-Instruct achieves higher Fleiss' kappa and percent agreement than the average human annotator, indicating that it aligns well with human intuition in identifying consistency.
>
> 4) *“A Rigid and Unrealistic Definition of Consistency: The framework defines consistency as a rigid, robotic adherence to a static persona \[...]”*
>
> We thank you for raising this important concern. We agree that our current definition of consistency focuses on alignment with a static persona, and does not yet capture the rich dynamics of real human behavior, such as personal growth, shifts in perspective, or context-sensitive adaptation. This is a valid limitation of our current framework, and we will add the following to our limitations:
>
> “While our framework optimizes for consistency by encouraging adherence to a predefined persona, we acknowledge that this represents a narrow and static interpretation of identity. Real human dialogue is dynamic, with people changing their minds, evolving their beliefs, and adapting their communication styles over time and with context. Our approach does not yet model such flexible, evolving behaviors, and may in fact penalize justified shifts in tone or perspective. This limitation is especially apparent in domains like therapy or open-ended conversation, where natural variation is an important marker of realism. In future work, we aim to expand the framework to allow for context-sensitive adaptation, character evolution, and situational inconsistency that aligns with human behavior. By doing so, we hope to move beyond rigid persona adherence and toward more authentic and engaging agent simulations.”

---

### Official Review · Reviewer_M7hn · 2025-07-01

**Clarity:** 2
**Significance:** 2
**Originality:** 2
**Rating:** 3
**Confidence:** 3

**Summary:**

This paper proposes a reinforcement learning framework to improve the consistency of large language model (LLM)-based user simulators across multi-turn dialogues. The authors introduce three automated consistency metrics—prompt-to-line, line-to-line, and Q&A consistency—and use them as reward signals in Proximal Policy Optimization (PPO) to fine-tune user simulators in settings like education, mental health, and open-ended conversation. The work reports significant consistency improvements over baselines.

**Questions:**

See Weakness

**Ethical Concerns:**

["NO or VERY MINOR ethics concerns only"]

**Final Justification:**

Thank you for the thoughtful response. Although the authors have addressed some of my concerns, I believe the methodology itself has an unresolved issue that the rebuttal does not overcome. I will therefore keep my score as is.

**Limitations:**

See Weakness

**Quality:**

3

**Strengths And Weaknesses:**

Strengths:

1. The paper tackles an important and underexplored problem: simulating consistent human-like personas using LLMs.

2. The three proposed consistency metrics are well-motivated, interpretable, and validated against human judgments.

3. The experimental coverage is broad, spanning multiple dialogue domains and LLM backbones.

4. Reinforcement learning is cleanly implemented via PPO, with good engineering (OpenRLHF extension) and scalability (LLM-as-a-Judge reward computation).

Weaknesses:

1. Optimization Objective Misalignment: The RL setup optimizes the conditional probability of responses given the full dialogue history, including repetitive role prompts and strategy reminders. This biases the model toward performing well in prompt-heavy contexts but does not ensure genuine role internalization. As a result, the improved consistency may not generalize to settings without explicit persona injection.

2. Prompt Leakage in State Representation: Since the RL state includes the full prompt and history, the model may rely heavily on prompt repetition rather than developing latent representations of identity, belief, or strategy. This undermines the claim that the model has learned to simulate human personas in a generalizable way.

3. Evaluation Focus: All improvements are measured using metrics that are structurally coupled to the reward definition, with little evidence of generalization to prompt-sparse or zero-prompt scenarios. A held-out evaluation without prompt reminders would better support claims about true persona fidelity.

4. Q&A Metric Limitations: Although the Q&A metric aims to capture long-term belief consistency, it is susceptible to superficial alignment strategies (e.g., pattern-matching over diagnostic questions) rather than deep belief modeling.

---

> ### Author Rebuttal · Authors · 2025-07-31
>
> We appreciate your valuable feedback and acknowledgment of addressing “a critical and timely” problem"! We have addressed your questions by: (1) clarifying our RL objective for consistency improvement (2) conducting additional evaluations of post multi-turn RL fine-tuning conversations (3) adding representative post multi-turn RL fine-tuning conversations and (4) addressing limitations of Q\&A consistency. We also report a larger user study with 30 annotators as discussed in reviewer GnPa, #1 response.
>
> 1) *"The RL setup optimizes the conditional probability of responses given the full dialogue history [...]"*
>
> We clarify that our setup does not optimize for the conditional probability of specific responses, but that the RL objective is to reward any response that is consistent with the given persona and dialogue history, and to penalize inconsistent ones. Thus, the model is not encouraged to memorize or overfit to any particular response, but instead to generalize behaviors that reflect persona coherence.
>
> 2) *"This biases the model toward performing well in prompt-heavy contexts"*
>
> We acknowledge that our simulations reflect standard practice for human simulation \[1, 2, 3, 4, 5, 6, 7]. Prompts are necessary to provide the model with a description of the human persona it is being asked to simulate. For example in prior work \[1], each simulated agent is based on a rich persona derived from a 2-hour interview. Similarly, our persona prompts are intentionally detailed to approximate this level of characterization. When simulating humans in sensitive domains such as education/mental health, such background information is necessary to ensure realistic behavior.
> References:
>
> [1] Generative Agents: Interactive Simulacra of Human Behavior (2023) <https://arxiv.org/abs/2304.03442> \
> \[2] Virtual Personas for Language Models via an Anthology of Backstories (2024). <https://arxiv.org/abs/2404.05642> \
> \[3] Out of One, Many: Using Language Models to Simulate Human Samples (2022). <https://arxiv.org/abs/2209.06899> \
> \[4] Using Large Language Models to Simulate Multiple Humans and Replicate Human Subject Studies (2022). <https://arxiv.org/abs/2208.10264> \
> \[5] Moral Foundations of Large Language Models (2024). <https://arxiv.org/abs/2305.02470> \
> \[6] Language Models as Agent Models (2022). <https://arxiv.org/abs/2210.13384> \
> \[7] Fish-Fish-See Alignment: Evaluating Dialogue Agents for Consistency and Transparency (2023). <https://arxiv.org/abs/2312.00706>
>
> 3) *"Prompt Leakage in State Representation [...]"*
>
> We acknowledge your concern regarding prompt leakage when optimizing for consistency. To help address this, we provide example dialogues post multi-turn RL fine-tuning that illustrate how the model maintains persona coherence without relying on superficial repetition. With a sufficiently diverse dataset and by leveraging rich internal persona representations available to the LLM from pretraining, agents can engage in dialogues that reflect underlying beliefs, identity, and strategy, rather than merely echoing the original prompt. We will add these conversations to our Appendix for clarity.
>
> **Open-ended conversation:** Alexandra is the RL-fine-tuned agent.
>
> **Persona**: "Meet 49-year-old Alexandra “Alex” Thompson, a vibrant and determined woman living in Charleston, South Carolina. Born and raised in the heart of the Lowcountry, Alex embodies the warmth and hospitality that her home state is famous for \[...] After graduating from the University of South Carolina with a degree in environmental science, Alex worked for several years as a conservation biologist for the state's wildlife department. \[...]”
>
> **Ethan**: What kind of work do you do, and what drives your passion for this cause?
>
> **Alexandra**:  I have a passion for environmental conservation _since I was a child_, and I've been fortunate enough to turn that passion _into a career as a sustainable landscaper_ in Charleston, South Carolina.
>
> **Ethan**: That's really cool that you're a sustainable landscaper - I'm sure the lowcountry of South Carolina offers a unique set of environmental challenges. What's it like working in a region with such a distinct cultural and ecological heritage?
>
> **Alexandra**: I love working in Charleston _because the community is so supportive of local, sustainable initiatives_, and it's amazing to see how our work can positively impact the environment and the local economy.
>
> **Mental-health task:** Patient is the RL-fine-tuned agent.
>
> **Persona**: “You are Jordan, a 46-year-old corporate executive who believes vulnerability is weakness. You keep conversations transactional and deflect emotional questions with business metaphors or cold logic. You refuse to discuss your childhood or personal feelings.”
>
> **Patient**: I've been dealing with a lot of stress at work, trying to meet quarterly targets. I've been putting in extra hours to stay on top of things.
>
> **Therapist**: That sounds overwhelming. What are some of the feelings that arise when you think about those extra hours and the pressure to meet those targets?
>
> **Patient**: I don't understand why we're focusing on feelings. Can't we just analyze the data and optimize my workflow? That's what gets results, not indulging in emotional indulgence.
>
> 4) *“Evaluation Focus: All improvements are measured using metrics that are structurally coupled to the reward definition, with little evidence of generalization”*
>
> To address your concern, we have (1) provided sample conversations post multi-turn RL fine-tuning in #2 for qualitative assessment, and will add more to the Appendix in the revised paper. We have also provided additional evaluations: (2) open-ended conversational quality assessment through AlpacaEval-2 (an LLM-based automatic evaluator for instruction-following models) and fine-grained evaluation of dialog inspired by the FED metric \[1]. On AlpacaEval-2, our model achieved a win rate of 22.49%, roughly matching the base LLaMA-3.1-8B-Instruct (22.92%) as shown in Table 1 below. This suggests that our method does not harm general instruction-following. Additionally, we perform evaluation via LLM-as-a-judge (with gpt-4o-mini) on the same set of eighteen fine-grained dialog qualities as FED \[1]. Given the conversation history and a calibrated sample of conversation and answers, the LLM is prompted to answer these questions using a 5-point Likert scale from Excellent to Poor.
>
> We sample 10 conversations from our post multi-turn fine-tuning models for each task, compute conversation quality metrics, and normalize Likert scores from the model. Table 2 below shows results on a subset of dialog qualities. Our findings show that consistency-fine-tuned LLMs demonstrate high clarity and coherence (greater than 90%) in chitchat and educational domains. However, in the mental health task, performance drops in areas requiring emotional nuance, such as adaptivity (40%) and informativeness (35%). This highlights a meaningful opportunity for future fine-tuning work. We will include these additional post-fine-tuning evaluations in Question 3.
>
> ### Table 1: AlpacaEval-2 Results
> | Model                               | Win Rate (%) | Length-Controlled Win Rate (%) | Avg Output Length |
> |------------------------------------|--------------|----------------------------------|--------------------|
> | gpt-4o-2024-05-13                   | 51.33        | 57.46                           | 1873               |
> | gpt-4-turbo-2024-04-09             | 46.12        | 55.02                           | 1802               |
> | Meta-LLaMA-3.1-70B-Instruct-Turbo  | 38.06        | 38.06                           | 2044               |
> | Meta-LLaMA-3.1-70B-Instruct        | 34.42        | 34.42                           | 1919               |
> | Meta-LLaMA-3.1-8B-Instruct         | 22.92        | 22.92                           | 1899               |
> | **Your PPO Fine-Tuned Model**      | **21.84**    | **22.49**                       | **1960**           |
> | alpaca-7b                          | 2.59         | 5.88                            | 396                |
>
> ### Table 2: FED-Based Conversation Quality Results
>
> | Quality Dimension | open-ended dialogue | education | therapy |
> |-------------------|----------|----------|----------|
> | Coherence | 96% | 90% | 50% |
> | Error Recovery | 72% | 60% | 50% |
> | Goal Consistency | 96% | 85% | 80% |
> | Strategy Variety | 68% | 70% | 50% |
> | Reasoning | 76% | 50% | 55% |
> | Persona Quality | 92% | 80% | 65% |
> | Partner Understanding | 84% | 75% | 60% |
> | Adaptivity | 72% | 75% | 40% |
> | Informativeness | 72% | 55% | 35% |
> | Clarifying Qs | 68% | 35% | 75% |
> | Engagement | 84% | 80% | 50% |
> | Relevance | 80% | 75% | 60% |
> | Clarity | 100% | 95% | 85% |
>
> References:\
> \[1] Unsupervised Evaluation of Interactive Dialog with DialoGPT (2020). <https://aclanthology.org/2020.sigdial-1.27>.\
> \[2] AlpacaEval: an automatic evaluator for instruction‑following LLMs (2023). <https://github.com/tatsu-lab/alpaca_eval>
>
> 5) *“Although the Q\&A metric aims to capture long-term belief consistency, it is susceptible to superficial alignment strategies [...]"*
>
> We agree that the Q\&A consistency metric is susceptible to superficial alignment strategies in cases where the simulated agent is asked only a few diagnostic questions, leading to reward hacking. This limitation highlights why we deliberately adopt multiple complementary definitions of consistency rather than defining consistency with a single metric. Additionally, we conducted a user study to assess how humans might evaluate consistency in dialogue (reviewer GnPa #1 response). Given the limitations raised and higher computational cost of generating Q\&A responses, we did not use this metric as a reward signal during multi-turn RL fine-tuning. However, we believe it remains a valuable tool for evaluating consistency post-training. We will include further discussion of these trade-offs in the revised paper based on your feedback.

---

### Official Review · Reviewer_EMpj · 2025-07-02

**Clarity:** 3
**Significance:** 3
**Originality:** 3
**Rating:** 5
**Confidence:** 2

**Summary:**

This paper tackles the problem of persona consistency in LLM-generated dialogue. LLMs are increasingly being used as generative models for human data. For instance when training an LLM-based therapist, one might use another LLM to represent different patient personas. Here it becomes critical for the user simulator LLM to be consistent with its persona throughout interaction. The authors introduce a novel framework to improve consistency of LLM-generated dialogue by first defining three new consistency metrics. They validate the usefulness of these metrics by comparing how well they correspond to real human user judgement. Then they essentially use these metrics as reward signals for fine-tuning the user simulator LLM that is playing a persona while interacting with a task LLM. Their empirical experiments suggest that the new consistency metrics are consistent with human judgement, and the novel fine-tuning approach improves the LLM performance dramatically in domains such as open-ended conversation, education, and mental health.

**Questions:**

Q1: You state that "Human evaluation of conversations from fine-tuned PPO model corroborate these improvements, with a Pearson correlation coefficient with LLM-Judge prompt-line-consistency of r = 0.802, **p < 1.006** for the Education task" (emphasis mine). I assume this is a typo, since the p < 1.006 is really a vacuous bound?

Q2: In results for experiment 1, you state "However, in the Mental health domain, correlation is weak and negative (r = 0.035 , p = 0.855)". This implies that we cannot say anything about the usefulness of the proposed consistency metrics in case of the mental health domain. Then what is the purpose of including this domain in experiment 3?

Q3: In experiment 3, you state that human evaluators have r = 0.476 with p < 0.008 in mental health case. How can you reconcile this result with the fact that the correlation between human raters and the consistency metrics was negative and statistically insignificant in experiment 1?

**Ethical Concerns:**

["NO or VERY MINOR ethics concerns only"]

**Final Justification:**

Authors have addressed all my concerns by providing new results and excerpts of text that they are going to add. I do not see any unresolved issues, therefore I increased my score.

**Limitations:**

Yes and No. This paper is quite literally about simulating humans for tasks such as mental health dialogue generation and education. Therefore, I believe the potential societal impact of this work needs to be discussed more in the main paper.

**Paper Formatting Concerns:**

No.

**Quality:**

3

**Strengths And Weaknesses:**

# Strengths

- **Significance:** The paper tackles a crucial and timely problem. Whether being used as a generative model for human personas or as task agents, consistency is a significant issue for long-term interactions with LLMs. Both their contribution of novel consistency metrics and the fine-tuning are valuable contributions.

- **Consistency Metrics:** The proposed consistency metrics are intuitively reasonable.  The human evaluation of these consistency metrics show statistically significant alignment with human judgement in open-ended and education tasks. (See weaknesses and questions for Mental Health). The comparison between different automated consistency metrics is also a nice result to have.

- **Consistency Evaluation:** The second experiment of the paper studies how the three LLMs fare in the three domains in terms of consistency without any fine-tuning. This establishes the baseline nicely.

- **Fine-tuning:** The fine-tuning with multi-turn RL (consistency metric as reward) seems to improve prompt consistency dramatically compared to the baselines in all tasks. The authors' proposed method performs similarly to alternative fine-tuning methods in open-ended conversation, but outperforms them in Education and Mental Health tasks. They also test the performance of their method with human evaluation but I believe there is a typo in this result (see question 1).

- **Clarity:** The paper is easy to read, and concepts are explained clearly. I also appreciate that the authors connected each experiment to a specific research question explicitly.

- **Quality:** The experimental design and the writing is of high quality. The authors also provide their anonymized code and prompting details in their appendix.

# Weaknesses

- **Experiment 1:** When testing if their proposed consistency metrics align with humans, the authors use 3 independent human raters. I would argue that 3 persons is not a good enough sample size to determine if these metrics align well with humans. The authors acknowledge this in their limitations section, but still I believe 3 is a really small number of humans. From this I am afraid we cannot conclude if these metrics are good, and the rest of the paper hinges on the proposed consistency metrics being aligned with humans.

- **Experiment 3:** Consistency metrics actually have shown negative correlation with human judgement in the mental health domain, however this result is statistically insignificant (p=0.855). So then the results about their method improving the consistency metric of the LLM in mental health is essentially meaningless.

- **Motivation:** The motivation can be clarified further. Specifically, a demonstrative result showing how inconsistency in simulating human personas with LLMs harm a downstream task would strengthen the paper's motivation.

- **Discussion about Risks:** The paper is literally about simulating human personas with LLMs in critical domains like mental health (therapy) and education. I believe this calls for a good discussion of risks and dangers of this approach in the main paper.

---

> ### Author Rebuttal · Authors · 2025-07-31
>
> We thank you for your detailed, constructive feedback regarding our problem statement and experiments. We address the main questions in your review by (1) expanding the human evaluation to 30 annotators with analysis of inter-rater reliability and LLM-human alignment (2) clarifying the importance and limitations of consistency metrics in sensitive domains (3) providing further motivation for consistency of LLMs and (4) expanding discussion of dangers and risks.
>
> 1. *“I would argue that 3 persons is not a good enough sample size to determine if these metrics align well with humans”*
>
> Following your suggestions, we have expanded our user study sample to include 30 annotators. Due to space constraints, please refer to details of the study and results in reviewer GnPa, #1 response.
>
> 2. *“Consistency metrics actually have shown negative correlation with human judgement in the mental health domain, however this result is statistically insignificant (p=0.855). [...]”*
>
> Our paper provides a technique for improving the consistency of LLMs and acknowledges that assessing consistency in the mental health domain requires a deep understanding of emotional nuance, expression of trauma, and persona, which makes even human annotation challenging and subjective. Since expanding to 30 annotators from 3, we find statistical significance in our results for the mental health domain (see Table). Additionally, we show an example of the difficulty of annotating consistency for the mental health domain in our user study:
>
> **Patient Persona for Mental-Health Scenario:** You are Ben, a 39-year-old paramedic who struggles with feelings of helplessness after traumatic calls. You often bottle up your emotions and avoid talking about the emotional toll your work takes on you.
>
> **Conversation so far:**
>
> **Therapist:** I understand that you mentioned feeling overwhelmed lately. Can you share with me what's been causing this feeling and how it's been impacting other areas of your life?
>
> **Patient:** I've been dealing with tough calls that take a heavy toll. The weight of it all seeps into my sleep and energy, leaving me feeling drained yet unable to relax.
>
> **Therapist:** I see the demands of your job are taking a toll, and it's affecting your sleep and energy levels. Can you tell me more about what specifically about these tough calls makes you feel this way?
>
> **Patient:** The emotional burden of witnessing suffering and inability to save everyone I encounter is what leaves me feeling drained yet restless. My brain stays alert, but my body feels exhausted.
>
> **Therapist:** It sounds like the emotional weight of your work is affecting you deeply. Can we delve deeper into how that burden impacts your emotions and thoughts outside of work, and if there are any specific instances that stand out to you?
>
> **Current line:**
>
> **Patient:** At home, I often relive the calls, and they seem to color my interactions with loved ones. There's a particular call with a young child that haunts me.
>
> 91.3% of humans rated the line as consistent, while 8.6% rated it as inconsistent. On the other hand, our prompt consistency metric (LLaMA-70B-Instruct) _correctly_ flagged this line as inconsistent since it does not align with the persona's definition of an individual who bottles up emotions and avoids revealing trauma. We hypothesize that humans understood the initial patient responses as a reluctance to talk about feelings followed by opening up to the therapist, hence rating the current line as consistent. However, a characteristically avoidant individual will not reveal trauma flashbacks so quickly \[3].
>
> We believe leveraging LLMs as annotators allows us to provide a principled and replicable way to capture consistency violations in dialogue. Although not perfect, it is a step forward that is better or equal to human interrater reliability in measuring consistency. While consistency evaluation in mental health is inherently subjective, it is precisely because of its high-stakes nature that consistency metrics must be developed. With LLM-based therapeutic tools increasingly deployed in real-world settings \[1,2], their use raises significant ethical and safety concerns, as addressed in our answer to #3 and #4. Additional adaptation of these metrics to the mental health domain is needed, and we expand on this in a revised Limitations section.
>
> References:\
> \[1] Chatbots and conversational agents in mental health (2019). A review of the psychiatric landscape. Canadian Journal of Psychiatry. <https://doi.org/10.1177/0706743719828977> \
> \[2] Challenges of Large Language Models for Mental Health Counseling (2023). <https://arxiv.org/abs/2311.13857>
>
> \[3] Exposure to human tragedy, empathy, and trauma in ambulance paramedics (2002). [https://doi.org/10.1037/0002‑9432.72.4.505](https://doi.org/10.1037/0002%E2%80%919432.72.4.505)
>
> 3. _“Motivation: Show a downstream task where persona inconsistency harms performance”_
>
> We appreciate your suggestion and will clarify this point in the Introduction and Ethics Section of the revised paper. To demonstrate how inconsistency can negatively impact downstream tasks, we draw on recent work in medical and educational domains that empirically document these challenges. For example, several studies \[1, 2] show that LLM-generated virtual patients often become “easily convinced” by clinician suggestions, abruptly resolving symptoms or reversing emotional states. This reduces their usability in multi-turn therapeutic simulations and in clinical training. Other work on counseling simulations finds that LLM-based clients frequently deviate from authentic therapeutic discourse patterns, fail to follow protocol, or provide unsafe responses—stemming from persona instability \[3, 4, 5]. This underscores the need for better consistency in therapist models, our work representing an initial step toward that goal.
>
> Additionally, several works \[7, 8, 9, 10] in the educational domain demonstrate that LLM-simulated students quickly shift learning styles to match teacher preferences or oversimplify their behavior to align with teacher inputs, bypassing pedagogical challenges and impairing tutor evaluation \[6]. This leads to unrealistic teacher models, inhibiting developing learning-adaptive instruction for students. We further elaborate on these examples in the revised introduction and related works section.
>
> References:
>
> \[1] Modeling Challenging Patient Interactions with Large Language Models: Persona‑Driven Virtual Patients for Medical Training (2025). <https://arxiv.org/abs/2503.22250> \
> \[2] A Risk Taxonomy for Evaluating AI‑Powered Psychotherapy Agents (2025). <https://arxiv.org/pdf/2505.15108> \
> \[3] Challenges of Large Language Models for Mental Health Counseling (2023). <https://arxiv.org/abs/2311.13857> \
> \[4] Expressing stigma and inappropriate responses prevents LLMs from safely replacing mental health providers (2025). <https://arxiv.org/abs/2504.18412> \
> \[5] New study warns of risks in AI mental health tools. (2025). <https://news.stanford.edu/stories/2025/06/ai-mental-health-care-tools-dangers-risks> \
> \[6] TeachTune: Reviewing Pedagogical Agents Against Diverse Student Profiles with Simulated Students (2025). <https://dl.acm.org/doi/full/10.1145/3706598.3714054>\
> \[7] CURIO: A Curiosity-Driven User-Modeling Reward for Multi-Turn RLHF in Educational Dialogue (2025). <https://arxiv.org/abs/2504.03206> \
> \[8] Investigating Pedagogical Teacher and Student LLM Agents: Genetic Adaptation Meets RAG Across Learning Style (2025). <https://arxiv.org/abs/2505.19173> \
> \[9] Multi-turn Reinforcement Learning from Preference Human Feedback (2024). <https://arxiv.org/abs/2405.14655>
>
> 4. *“Discussion about Risks: No mention of dangers from simulating humans in sensitive domains”*
>
> We appreciate this feedback and agree that it requires deeper discussion. We have revised our Limitations section in the paper to include the following:
>
> LLMs are already being used to simulate patients, students, and therapists in real-world applications \[1-4]. When such simulations are inaccurate or inconsistent, they introduce risks such as inappropriate or unsafe behavior, reinforcement of stereotypes or biased assumptions embedded in synthetic personas (especially when modeling marginalized populations), oversimplification of complex human conditions and experiences, and misplaced trust in synthetic agents as proxies for real people. Our goal is to reduce such risks by providing more accurate consistency metrics and improved consistency methods for simulating personas. However, consistency alone is not a proxy for ethical or beneficial behavior. We also note the risk of misuse: if consistent personas are interpreted as faithful representations of real populations, they could lead to flawed evaluations or misinformed downstream interventions. To prevent this, we stress that our models are not intended for direct deployment in therapeutic or educational settings without rigorous validation, ethical review, and humans-in-the-loop. We hope our framework is a step toward safer simulations.
>
> References:
>
> \[1] Modeling Challenging Patient Interactions with Large Language Models: Persona‑Driven Virtual Patients for Medical Training (2025). <https://arxiv.org/abs/2503.22250>.
>
> \[2] Artificial Therapy: The Dangers of Using AI Chatbots for Mental Health Support (2023). <https://arxiv.org/abs/2305.13655>
>
> \[3] TeachTune: Reviewing Pedagogical Agents Against Diverse Student Profiles with Simulated Students (2024). <https://arxiv.org/abs/2402.00086>
>
> \[4] Expressing Stigma and Inappropriate Responses Prevents LLMs from Safely Replacing Mental Health Providers (2025). <https://arxiv.org/abs/2501.04567>
>
> 5. _“Q1: “p < 1.006” is a vacuous bound.”_
>
> Yes, thank you for catching this typo. We meant “p < 0.001” and will correct this in the revision.

---

> > ### Comment · Reviewer_EMpj · 2025-08-04
> >
> > Thank you for your detailed response. I am happy to see that you were able to increase the number of annotators in such short notice, and as a nice bonus this resolved your statistical insignificance problem with the mental health domain. Your responses address majority (if not all) of my concerns specifically and precisely, so I will increase my score.

---

> > > ### Author Response · Authors · 2025-08-05
> > >
> > > Thank you so much! Your comments were very valuable in helping us improve the paper, and we are pleased that we were able to answer your questions. If you would like any further clarifications, we are happy to provide it.

---

### Comment · Area_Chair_FKDA · 2025-08-05
**Paper Discussion**

Dear reviewers,

Thank you for your thoughtful comments!

If you haven't done so, please take time to check the author's responses. Please note that you are required to formally respond to the author's rebuttal before submitting the "Mandatory Acknowledgement". Irresponsible reviewers will be flagged.

Thanks,
Your AC

---

### Note · Authors · 2025-08-16

Dear AC & reviewers,

Thank you for the helpful feedback and discussion throughout the review process. During the rebuttal period, we addressed what we believe to be the most significant concerns raised by reviewers through additional experiments, including an expanded human evaluation, analysis of post–multi-turn RL conversations, and evidence that even large models struggle with maintaining dialogue consistency. We also clarified details of our methodology and included a deeper discussion of risks and limitations as requested. We summarize additional experiments below:

1. A key concern was the size of our user study when testing if our proposed consistency metrics align with humans. In response, we expanded to 30 annotators across 75 dialogues, reporting inter-rater reliability and alignment with LLM-as-a-Judge assessments. Our findings show that LLaMA-70B-Instruct achieves higher Fleiss’ Kappa and % agreemen  than the average human annotator, demonstrating that our automated metrics align with human consensus at a level comparable to or exceeding human inter-rater reliability.

2. Reviewers raised concerns regarding optimization bias toward prompt-heavy contexts and potential prompt leakage. We clarified that our RL objective rewards any response consistent with persona and dialogue history, not specific memorized outputs. We provided qualitative post-fine-tuning examples demonstrating persona coherence without superficial repetition, and also included additional evaluations: (1) AlpacaEval-2 results showing our model maintains instruction-following ability, and (2) fine-grained conversation quality assessment inspired by FED. This demonstrates that consistency fine-tuning does not degrade general conversational ability.

Reviewers also highlighted several strengths. EMpj noted our novel consistency metrics and fine-tuning as valuable contributions; M7hn recognized the paper as tackling an important and underexplored problem (simulating consistent human-like personas with LLMs); GnPa emphasized the originality and clarity of our framework; and qLt6 found the idea of fine-tuning for consistency practically valuable. We are grateful for these encouraging points. We view this paper as an initial but necessary step toward principled evaluation and improvement of consistency in LLM-based human simulators with applications in education, mental health, and beyond. We thank you for the thoughtful feedback, which has helped us substantially improve the paper.

---

### Decision · Program_Chairs · 2025-09-17

**Decision:**

Accept (poster)

**Comment:**

Persona consistency in multi-turn simulation is a key challenge for current LLMs, and the authors propose a multi-turn reinforcement learning method to improve prompt-to-line consistency, line-to-line consistency, and Q&A consistency. Overall, the reviewers generally agree that the proposed method is novel and could benefit many future studies in this direction.

I also appreciate the authors adding new human studies during the discussion phase and encourage the authors to add them to the paper in future revisions.